# DISTRIBUTIONAL STRUCTURED PRUNING BY LOWER BOUNDING THE TOTAL VARIATION DISTANCE USING WITNESS FUNCTIONS

## ABSTRACT

Recent literature introduced the notion of distributional structured pruning (DSP) in Deep Neural Networks by retaining discriminative filters that can effectively differentiate between classes. Crucial to DSP is the ability to estimate the discriminative ability of a filter, which is defined by the minimum pairwise Total Variation (TV) distance between the class-conditional feature distributions. Since the computation of TV distance is generally intractable, existing literature assumes the class-conditional feature distributions are Gaussian, thereby enabling the use of the tractable Hellinger lower bound to estimate discriminative ability. However, the Gaussian assumption is not only restrictive but also does not typically hold. In this work, we address this gap by deriving a lower bound on TV Distance which depends only on the moments of witness functions. Using linear witness functions, the bound establishes new relationships between the TV Distance and well-known discriminant-based classifiers, such as Fisher Discriminants and Minimax Probability machines. The lower bounds are used to produce a variety of pruning algorithms called WitnessPrune by varying the choice of witness function. We empirically show that we can achieve up to 7% greater accuracy for similar sparsity in hard-to-prune layers using a polynomial witness function as compared to the state-of-the-art.[1]

# 1 INTRODUCTION

The size of modern Deep Neural Networks (DNNs), some of which possess billions of parameters, makes deploying them in real-world scenarios a challenging problem (Hoefler et al., 2021; Prakash et al., 2019; Molchanov et al., 2019). This has necessitated research into model compression techniques, by which model sizes are reduced in order to satisfy real-world performance requirements, such as inference time and power consumption. A variety of methods have been developed, including quantization (Gholami et al., 2022), knowledge distillation (Gou et al., 2021), hashing (Deng et al., 2020), and pruning (Hoefler et al., 2021; Blalock et al., 2020). Structured pruning - processes by which entire filters or neurons are removed from DNNs - is a popular and effective tool for improving real-world performance in terms of inference time and power consumption of DNNs, without requiring any additional hardware and software (Hoefler et al., 2021).

However, most contemporary methods for structured pruning require access to the training data or the loss function, which may be unavailable due to constraints such as privacy and security (Yin et al., 2020; Narshana et al., 2022), making the development of algorithms to prune models in this setting an active area of research. In this work, we relax this constraint, and address the problem of structured pruning without training set or loss function access, but with access to the distribution of the data, such as natural samples or moments.

A new approach for solving this problem was proposed in Murti et al. (2022), which identified *discriminative filters* - filters that generate features capable of effectively discriminating between classes - and pruning non-discriminative filters. This paradigm of pruning is called *distributional pruning*, as it requires some access to the class-conditional distributions.

---

[1]Our code is available at the anonymous repository at `https://shorturl.at/kmrE9`

The key challenge in distributional pruning lies in quantifying discriminative ability. In Murti et al. (2022), the class conditional features are assumed to be Gaussian distributed, and the discriminative ability is quantified by computing the Hellinger-lower bound on the Total Variation (TV) distance between Gaussians. This assumption may be unrealistic, as our experiments indicate that class-conditional distributions are in general not Gaussian. However, estimating the TV distance is known to be intractable (Bhattacharyya et al., 2022). Recent works have addressed finding lower bounds on the TV distance between known distributions, such as (Hardt & Price, 2015; Davies et al., 2022). To address this problem, we adopt a *witness function* based approach to lower bounding the TV distance.

In this work, we address the problem of identifying and pruning non-discriminative filters without assuming prior knowledge about the class conditional feature distributions. To do so, we develop witness function based lower bounds on the TV distance, with which we then derive a variety of distributional pruning algorithms. We state our contributions formally below.

- We produce the empirical observation that the class conditional distributions of feature maps are, in general, not Gaussian in section 7. This motivates us to propose a novel witness function-based lower bound on the TV distance between distributions; this result is stated in Theorem 1. The bounds require no prior knowledge of the distributions, apart from the boundedness of the distributions moments. Motivated by the use of the Hellinger distance in contemporary work on distributional pruning (Murti et al., 2022), we also apply our strategy to derive witness function-based lower bounds on the Hellinger distance as well. This result is presented in Theorem 2.

- Using a careful choice of witness function, our bounds reveal new connections between discriminant-based classifiers like the Fisher linear discriminant and the Minimax Probability Machine, and the TV and Hellinger distances. In particular, in Corollary 1, we show that the TV distance is lower bounded by a monotonic function of the Fisher discriminant, and in Corollary 2, we show that the TV distance is lower bounded by monotonic functions of the Minimax Probability Machine.

- Using these lower bounds, we derive a family of algorithms for distributional structured pruning, called WITNESSPRUNE(stated in Algorithm 1). WITNESSPRUNE uses the lower bound proposed in Theorem 1 to quantify the discriminative ability of filters, while only assuming the moments of the feature distributions are bounded. By varying the choice of witness function, we obtain a variety of distributional pruning algorithms.

- We produce a slate of experiments to illustrate the efficacy of our method. First, we show that our witness function-based lower bounds can effectively compute a lower bound on the TV distance even when the means are not well separated; we illustrate this on the Two Spirals dataset, and a pair of zero mean Gaussians. Motivated by these results, we compare the efficacy of WITNESSPRUNE on pruning hard-to-prune layers, and show that WITNESSPRUNE consistently achieves up to 6% higher accuracies than TVSPrune when pruning hard-to-prune layers. We then compare the efficacy of WITNESSPRUNE for pruning VGG nets and ResNets trained on CIFAR10, and show that we achieve up to 76% sparsity with a .12% reduction in accuracy on VGG19.

## 2 BACKGROUND AND NOTATION

In this section, we introduce our notation, and provide basic background definitions. For an integer $N > 0$, let $[N] := \{1, \cdots, N\}$. Let $\mathbf{0}_N$ be a vector of zeros of dimension $N$. Let $\text{sort}_B(\{a_1, \cdots, a_M\})$ be the set of the $B$ largest elements of $\{a_1, \cdots, a_M\}$. Suppose $\mathbb{P}$, $\mathbb{Q}$ are two distributions supported on $\mathcal{X}$, with densities given by $p(x)$ and $q(x)$. For a function $f : \mathcal{X} \to \mathbb{R}$, let $\bar{f}_p = \mathbb{E}_{x \sim \mathbb{P}}[f(x)]$, and let $\bar{f}_p^{(2)} = \mathbb{E}_{x \sim \mathbb{P}}[f(x)^2]$. Let $\mathcal{D}$ be a data distribution. Suppose the dataset has $C$ classes, then let $\mathcal{D}_c$ be the class-conditional distribution of the $c$-th class, and let $\mathcal{D}_{\bar{c}}$ be the distribution of the complement of $c$ (that is, samples are drawn from all classes other than $c$).

Suppose we have a neural network $\mathcal{W} = (W_1, \cdots, W_L)$. Each layer yields (flattened) representations

$$Y^l(x) = \left[ Y_1^l(X), \cdots, Y_{N_l}^l(X) \right], \tag{1}$$

where $N_l$ is the number of filters in layer $l$. Since $Y^l$ is dependent on $X$, we assume that $Y^l(X) \sim \mathcal{D}^l$, and $Y_j^l(X) \sim \mathcal{D}_j^l$. Furthermore, let $\mathcal{D}_{j,c}^l$ and $\mathcal{D}_{j,\bar{c}}^l$ be the class-conditional distributions and class-complement distributions of $Y_j^l(X)$ respectively.

Next, we define the Fisher Linear Discriminant and the Minimax Probability Machine Lanckriet et al. (2001).

**Definition 1.** *Let $\mathbb{P}$ and $\mathbb{Q}$ be distributions supported on $\mathcal{X}$, with moments $\mu_p, \Sigma_p$ and $\mu_q, \Sigma_q$. Then,*

$$\mathsf{Fish}(\mathbb{P},\mathbb{Q};u) = \frac{\left(u^\top(\mu_p - \mu_q)\right)^2}{u^\top(\Sigma_p + \Sigma_2)u} \quad and \quad \mathsf{MPM}(\mathbb{P},\mathbb{Q};u) = \frac{|u^\top(\mu_p - \mu_q)|}{\sqrt{u^\top\Sigma_p u} + \sqrt{u^\top\Sigma_q u}}. \quad (2)$$

*If we choose the optimal $u$, denoted by $u^*$, we write $\mathsf{Fish}(\mathbb{P},\mathbb{Q};u^*) = \mathsf{Fish}(\mathbb{P},\mathbb{Q})^*$ and $\mathsf{MPM}(\mathbb{P},\mathbb{Q};u^*) = \mathsf{MPM}(\mathbb{P},\mathbb{Q})^*$.*

We define the TV and Hellinger distances as follows.

**Definition 2.** *Let $\mathbb{P}$ and $\mathbb{Q}$ be two probability measures supported on $X$, and let $p$ and $q$ be the corresponding densities. Then, we define the Total Variation Distance $\mathrm{TV}$ and the Squared Hellinger distance $\mathrm{H}^2$ as*

$$\mathrm{TV}(\mathbb{P},\mathbb{Q}) = \sup_{A \subset X} |\mathbb{P}(A) - \mathbb{Q}(A)| = \frac{1}{2}\int_X |p(x) - q(x)|\,dx$$

$$\mathrm{H}^2(\mathbb{P},\mathbb{Q}) = \frac{1}{2}\int_X \left(\sqrt{p(x)} - \sqrt{q(x)}\right)^2 dx$$

## 3 REVIEW OF DISTRIBUTIONAL STRUCTURED PRUNING

In this section, we set up the problem of structured pruning, and formalize the notion of distributional pruning in terms of the performance of a Bayes optimal classifier trained on the generated features.

**Structured Pruning** In this section, we review structured pruning. Structured pruning, as defined in Hoefler et al. (2021); Blalock et al. (2020), is the removal of entire filters from a convolutional neural network. We aim to find the most sparse model that satisfies a sparsity budget. A large body of work exists that explores this problem, as noted in Hoefler et al. (2021) and the references therein In this work, we solve this problem in a layer-wise fashion.

**Distributional Pruning** In this section, we describe how distributional information about the feature maps can be used for structured pruning. In Murti et al. (2022), this was accomplished by identifying *discriminative filters*, which are filters that generate features with $\min_{c,c' \in C} \mathrm{TV}(\mathcal{D}_{j,c}^l, \mathcal{D}_{j,c'}^l)$ that are small, and pruning them. However, as mentioned previously, estimating lower bounds on the TV distance between arbitrary distributions, given moments, remains a challenging task. In Murti et al. (2022), this challenge is overcome by assuming the class conditional distributions are spherical Gaussians; that is, $Y_j^l(X) \sim \mathcal{N}(\mu_{j,c}^l, (\sigma_{l,c}^l)^2 I) \equiv \mathcal{D}_{j,c}^l$. Using this assumption enables the use of the Hellinger lower bound on the TV distance as follows.

$$\mathrm{TV}(\mathcal{D}_{j,c}^l, \mathcal{D}_{j,c'}^l) \geq \mathrm{H}^2(\mathcal{D}_{j,c}^l, \mathcal{D}_{j,c'}^l) = 1 - \left(\frac{2\sigma_{j,c}^l \sigma_{j,c'}^l}{(\sigma_{j,c'}^l)^2 + (\sigma_{j,c}^l)^2}\right)^{\frac{d}{2}} e^{-\frac{\Delta}{4}} \geq 1 - e^{-\frac{\Delta}{4}}$$

where $\Delta = \mathsf{Fish}^*(\mathcal{D}_{j,c}^l, \mathcal{D}_{j,c'}^l)$. Using this bound, the TVSPrune algorithm proposed in Murti et al. (2022) checks whether $1 - e^{-\Delta/4}$ is small for each filter, and if so, removes them.

**Discussion on TVSPrune:** While TVSPrune is a powerful pruning algorithm, it has a notable drawback. Our experiments show that the class conditional distributions are either non-Gaussian or in hard-to-prune layers, have poorly separated means. This undercuts the fundamental assumptions laid out in Murti et al. (2022). This motivates the following questions.

1. Since the class conditional filter outputs are either non-Gaussian or poorly separated (as noted in Murti et al. (2022), can we derive tractable lower bounds on the TV distance between arbitrary distributions provided sufficient moment information?

2. Can these lower bounds be applied to the problem of structured pruning, and reveal new insights into distributional pruning?

# 4 WITNESS FUNCTION-BASED LOWER BOUNDS FOR THE TOTAL VARIATION DISTANCE

In this section, we derive a lower bound on the Total Variation Distance that relies on the moments of a *witness function*, a scalar-valued function whose moments can be used to derive bounds on divergences between distributions. We then adapt this lower bound to a variety of scenarios, depending on the extent of the information about the distributions available to us. When access to only the first two moments is available, we derive lower bounds on the total variation distance based on the Fisher linear discriminant and the minimax probability machine.

## 4.1 LOWER BOUNDS ON THE TV AND HELLINGER DISTANCES

We derive robust lower bounds on the TV and Hellinger distances that are functions of the moments of a witness function $f : \mathcal{X} \to \mathbb{R}$, where $\mathcal{X}$ is the support of the distribution(s) in question. We begin by defining a metric between sets of distributions.

Estimating the Total Variation is known to be #P complete (Bhattacharyya et al., 2022). Estimating lower bounds on the TV distance is an active area of research (see Davies et al. (2022) and the references within), with a variety of applications from clustering (Bakshi & Kothari, 2020; Hardt & Price, 2015) to analyzing neural networks (Yu et al., 2018). However, most bounds such as those presented in Davies et al. (2022) require prior knowledge about the distributions, and tractable estimation of lower bounds given access to collections of moments or samples, without assumptions on the distributions themselves, remains an open problem.

Let $\mathcal{S}_k(\mathbf{P}) := \left\{ \mathbb{P} \; : \; \mathbb{E}_{X \sim \mathbb{P}} \left[ X_1^{d_1} \cdots X_n^{d_n} \right] = \mathbf{P}_{d_1 \cdots d_n}, \; \sum_i d_i \leq k \right\}$ be the set of probability measures whose moments are given by $\mathbf{P}$, where $\mathbb{E}_{X \sim \mathbb{P}} \left[ X_1^{d_1} \cdots X_n^{d_n} \right] = \mathbf{P}_{d_1 \cdots d_n}$; similarly, let $\mathcal{S}_k(\mathbf{Q})$ be the set of measures whose moments are given by $\mathbf{Q}$. For any random variable $X \in \mathbb{R}^d$ supported on $\mathcal{X}$, suppose $\varphi : \mathbb{R}^d \to \mathbb{R}^n$ for which there exist functions $g$ and $G$ such that $\mathbb{E}_X[\varphi(X)] = g(\mathbf{P})$ and $\mathbb{E}\left[\varphi(X)\varphi(X)^\top\right] = G(\mathbf{P})$. Given two collections of moments of the same order, we want to measure the worst-case TV separation between *all* distributions possessing the moments given in $\mathbf{P}$ and $\mathbf{Q}$.

**Theorem 1.** *Suppose $\mathbf{P}$ and $\mathbf{Q}$ are sets of moments of two probability measures supported on $\mathcal{X}$. Let $f = u^\top \varphi(X)$, be a witness function, with $\bar{f}_p^{(2)} = u^\top g(\mathbf{P})$, and $\bar{f}_p^{(2)} = u^\top G(\mathbf{P})u$. Then, for any $\mathbb{P} \in \mathcal{S}_k(\mathbf{P})$, $\mathbb{Q} \in \mathcal{S}_k(\mathbf{Q})$, supported on a set $\mathcal{X} \subseteq \mathbb{R}^d$, we have*

$$D_{\mathrm{TV}}(\mathcal{S}_k(\mathbf{P}), \mathcal{S}_k(\mathbf{Q})) = \min_{\mathbb{P} \in \mathcal{S}_k(\mathbf{P}), \mathbb{Q} \in \mathcal{S}_k(\mathbf{Q})} \mathrm{TV}(\mathbb{P}, \mathbb{Q}) \geq \sup_{u \in \mathbb{R}^n} \frac{\left(u^\top (g(\mathbf{P}) - g(\mathbf{Q}))\right)^2}{2u^\top (G(\mathbf{P}) + G(\mathbf{Q}))u} \quad (3)$$

*Proof Sketch.* We provide a sketch of the proof. We express the quantity $f_p - f_q$ in terms of the densities $p(X)$ and $q(X)$. We then isolate the integral of $|p(x) - q(x)|$. After rearranging terms, we obtain the result. For the full proof, we refer readers to Appendix A. □

Theorem 1 is a worst-case lower bound on the TV distance between distributions with given moments. Note that it is straightforward to extend this result to the case where the moments of witness functions match instead. While we focus our results on the case where $f(x) = u^\top \varphi(x)$, where $\varphi(x)$ is a vector of monomials, this bound is valid for any choice of $f$ with bounded first and second moments.

Next, we utilize the same methodology to produce a lower bound on the squared Hellinger distance.

**Theorem 2.** *Suppose $\mathbf{P}$ and $\mathbf{Q}$ are sets of moments of two probability measures supported on $\mathcal{X}$. Let $f = u^\top \varphi(X)$, be a witness function, with $\bar{f}_p^{(2)} = u^\top g(\mathbf{P})$, and $\bar{f}_p^{(2)} = u^\top G(\mathbf{P})u$. Then, for any $\mathbb{P} \in \mathcal{S}_k(\mathbf{P})$, $\mathbb{Q} \in \mathcal{S}_k(\mathbf{Q})$, supported on a set $\mathcal{X} \subseteq \mathbb{R}^d$, we have*

$$D_{\mathrm{H}}(\mathcal{S}_k(\mathbf{P}), \mathcal{S}_k(\mathbf{Q})) = \min_{\mathbb{P} \in \mathcal{S}_k(\mathbf{P}), \mathbb{Q} \in \mathcal{S}_k(\mathbf{Q})} \mathrm{H}(\mathbb{P}, \mathbb{Q})^2 \geq \sup_{u \in \mathbb{R}^n} \frac{2\left(u^\top (g(\mathbf{P}) - g(\mathbf{Q}))\right)^2}{\left(\sqrt{u^\top G(\mathbf{P})u} + \sqrt{u^\top G(\mathbf{Q})u}\right)^2} \quad (4)$$

The proof for this theorem is similar to that of Theorem 1, and is presented in full in Appendix A.

## 4.2 CONNECTIONS TO DISCRIMINANT BASED CLASSIFIERS

In this section, we exploit the bound stated in Theorem 1 to reveal extensive connections between the total variation distance and discriminant-based linear classifiers, specifically the Fisher Linear Discriminant and the Minimax Probability Machine.

**Connection to the Fisher Discriminant** In this section, we leverage the results of Theorem 1 to illustrate the connection between the Total Variation distance between two distributions, and their Fisher Discriminant. Specifically, we show that the TV distance is lower-bounded by a monotonic function of the Fisher Discriminant. We state this result formally in Corollary 2.

**Corollary 1.** *Let $\mathbb{P}, \mathbb{Q}$ be two probability measures supported on $X \subseteq \mathbb{R}^d$, let $p$ and $q$ be the corresponding densities, and let $\mu_p$, $\mu_q$ and $\Sigma_p$, $\Sigma_q$ be the means and variances of $\mathbb{P}$ and $\mathbb{Q}$ respectively. Then,*

$$\mathrm{TV}(\mathbb{P}, \mathbb{Q}) \geq \frac{\mathsf{Fish}^*(\mathbb{P}, \mathbb{Q})}{2 + \mathsf{Fish}^*(\mathbb{P}, \mathbb{Q})}. \tag{5}$$

This lower bound can be improved upon by selecting a witness function of the form $f(x; u) = u^\top \varphi(x)$ where $\varphi(x)$ is a vector of basis functions (such as monomials, if $f(x; u)$ is a polynomial).

**Connection to Minimax Probability Machine** In this section, we provide lower bounds on the Total Variation and Hellinger distances between the two distributions in terms of the minimax probability machine.

**Corollary 2.** *Let $\mathbb{P}, \mathbb{Q}$ be two probability measures supported on $X \subseteq \mathbb{R}^d$, let $p$ and $q$ be the corresponding densities, and let $\mu_p$, $\mu_q$ and $\Sigma_p$, $\Sigma_q$ be the means and variances of $\mathbb{P}$ and $\mathbb{Q}$ respectively. Then,*

$$\sqrt{\mathrm{TV}(\mathbb{P}, \mathbb{Q})} \geq \frac{\mathsf{MPM}^*(\mathbb{P}, \mathbb{Q})}{\sqrt{2} + \mathsf{MPM}^*(\mathbb{P}, \mathbb{Q})} \tag{6}$$

We present the proof for this Corollary in Appendix B. As with Corollary 1, the lower bound can be improved by choosing $f(x) = u^\top \varphi(x)$, and is also amenable to kernelization.

## 5 WITNESSPRUNE- ALGORITHMS FOR DISTRIBUTIONAL PRUNING

In this section, we leverage the lower bounds proposed in Theorem 1 and Corollaries 1 and 2 to develop WITNESSPRUNE, a one-shot pruning algorithm that requires no access to the training data or loss function, but only access to the data distributions. WITNESSPRUNE aims to identify and prune the least discriminative filters.

### 5.1 DISTRIBUTIONAL PRUNING WITH WITNESSPRUNE

A key drawback of the approach proposed in Murti et al. (2022) is that the class-conditional feature distributions are assumed to be Gaussian, which is an impractical assumption. Furthermore, by assuming the distributions are Gaussian and using the closed-form Hellinger lower bound, $\binom{C}{2}$ pairwise TV distances need to be computed for each filter. We now derive the WITNESSPRUNEalgorithm.

Let $Y^l(X)$ be the features generated by layer $l$ of a neural network as defined in equation 1. We choose a test function $f = u^\top \varphi(Y_j^l(X)) = \varphi_j^l(X)$, and let $\bar{f}_{l,j,c}(u) = \mathbb{E}_{X \sim \mathcal{D}_c}[u^\top \varphi_j^l(X)]$, $\bar{f}_{l,j,\bar{c}}(u) = \mathbb{E}_{X \sim \mathcal{D}_{\bar{c}}}[u^\top \varphi_j^l(X)]$, $\bar{f}_{l,j,c}^{(2)}(u) = \mathbb{E}_{X \sim \mathcal{D}_c}[(u^\top \varphi_j^l(X))^2]$ and $\bar{f}_{l,j,\bar{c}}^{(2)}(u) = \mathbb{E}_{X \sim \mathcal{D}_{\bar{c}}}[(u^\top \varphi_j^l(X))^2]$. Next, define $r_j^l$ to be the saliency score for the $j$th filter in the $l$th layer as

$$r_j^l = \min_{c \in [C]} \max_u \frac{\left(\bar{f}_{l,j,c}(u) - \bar{f}_{l,j,\bar{c}}(u)\right)^2}{\bar{f}_{l,j,c}^{(2)}(u) + \bar{f}_{l,j,\bar{c}}^{(2)}(u)}. \tag{7}$$

We use the lower bound established in Theorem 1 on the TV distances between the class conditional distributions to measure the *saliency* or importance

of a given filter. With this, we formally state WITNESSPRUNE in Algorithm 1.

---

**Algorithm 1:** WITNESSPRUNE

---

**Input:** Class conditional distributions $\mathcal{D}_c$, $c \in [C]$, Pretrained CNN with parameters
  $\mathcal{W} = (W_1, \cdots, W_L)$, layerwise sparsity budgets $B^l$, witness function $f$

**for** $l \in [L]$ **do**
 Set $S^l = [s_1^l, \cdots, s_{N_l}^l] = \mathbf{0}_{N_l}$
 Compute $r_j^l$ using equation 7 for all $j$.
 **if** $j \in \text{sort}_{B_l}(\{r_j^l\}_{j=1}^{N_l})$ **then**
  Set $s_j^l = 1$

**Output:** Binary masks $S^1, \cdots, S^L$,
**return** $\hat{\mathcal{W}}$

---

The WITNESSPRUNE algorithm has several advantages. First, as compared to TVSPrune, it requires that only $C$ TV distances be computed at each step. Second, by varying the choice of witness function, we obtain new algorithms for structured pruning; we can choose different witness functions for each class as well.

## 5.2 VARIANTS OF WITNESSPRUNE

In this section, detail specific variants of the WITNESSPRUNE algorithm. We describe these variants of WITNESSPRUNE in greater detail in Appendix C.

**Linear Witness Functions:** By choosing $f_c(x) = u^\top(x - \frac{\mu_c - \mu_{\bar{c}}}{2})$, we recover the WITNESSPRUNE-F and WITNESSPRUNE-M pruning algorithms. Thus, we would select $r_j^l = \min_{c \in C} \text{Fish}^*(\mathcal{D}_{j,c}^l, \mathcal{D}_{j,\bar{c}}^l)$ for WITNESSPRUNE-F, and $r_j^l = \min_{c \in C} \text{MPM}^*(\mathcal{D}_{j,c}^l, \mathcal{D}_{j,\bar{c}}^l)$ for WITNESSPRUNE-M. We can make this choice as is since the lower bounds on the TV distance obtained in Corollaries 1 and 2 are monotonic functions of the Fisher discriminant and the minimax probability machine.

**Nonlinear Witness Functions** By choosing nonlinear witness functions of the form $f(x) = u^\top(\phi(x) - \frac{\bar{\phi}_c + \bar{\phi}_{\bar{c}}}{2})$, where $\bar{\phi}_c = \mathbb{E}_{x \sim \mathcal{D}_c}[\phi(x)]$, we can employ the same tools as with the case of linear witness functions. However, the variants of WITNESSPRUNE-F and WITNESSPRUNE-M are better able to measure discriminative ability, and achieve a higher lower bound on TV distance. In practice, we choose $\phi(x)$ to be a quadratic function - yielding the algorithms WITNESSPRUNE-FQ and WITNESSPRUNE-MQ - though any choice of function with bounded moments with respect to the class-conditional distributions would be applicable.

**Ensembles of Witness Functions** To improve the effectiveness of WITNESSPRUNE, we can choose the largest possible measure of discriminative ability we can choose $r_j^l$ to be the largest of several computed lower bounds. For instance, if we consider linear witness functions, we can choose
$$r_j^l = \min_{c \in C} \max\left\{ \frac{\text{Fish}^*(\mathcal{D}_{j,c}^l, \mathcal{D}_{j,\bar{c}}^l)}{\text{Fish}^*(\mathcal{D}_{j,c}^l, \mathcal{D}_{j,\bar{c}}^l) + 2}, \left(\frac{\text{MPM}^*(\mathcal{D}_{j,c}^l, \mathcal{D}_{j,\bar{c}}^l)}{\text{MPM}^*(\mathcal{D}_{j,c}^l, \mathcal{D}_{j,\bar{c}}^l) + \sqrt{2}}\right)^2 \right\}.$$ We call this variant of the algorithm WITNESSPRUNE-E. Unlike with WITNESSPRUNE-F or WITNESSPRUNE-M, we cannot directly use the discriminant functions as saliencies, as the scale of the discriminants can be different. Thus, we use the lower bounds on TV distance directly.

**Using the BatchNorm random variables** We can also measure the discriminative ability of filters in terms of the TV distance between the class conditional BatchNorm (BN) random variables. Given a filter that generates a feature map $Y_j^l(X)$, we measure the TV distance between the class conditional distributions of the random variable $\text{BN}_j^l(X) = 1^\top Y_j^l(X)$. This choice is motivated by work such as Yin et al. (2020), which establishes that significant distributional information is stored in the BN moments. Measuring the TV distance between the class-conditional distributions of $\text{BN}_j^l(X)$ serves as an effective means of measuring the discriminative ability of filters, and provides significant improvements in storage overhead.

## 6 A DISCRIMINATION PERSPECTIVE ON DISTRIBUTIONAL PRUNING

In this section, we offer a new perspective on distributional pruning. Our algorithm WIT-NESSPRUNE identifies filters that generate features whose class conditional distributions are poorly separated in terms of the TV distance. Since the TV distance and the misclassification probability of the Bayes optimal classifier are connected by the identity $2R^*(\mathbb{P}, \mathbb{Q}) = 1 - \text{TV}(\mathbb{P}, \mathbb{Q})$, we can reformulate distributional pruning as identifying those filters that generate features for which the Bayes optimal classifier has high error, and pruning them. In this spirit, a natural extension of our results is to derive bounds on the generalization error of the Fisher and Minimax classifiers in terms of the Bayes error of the data distributions, and vice versa. To the best of our knowledge, these are the first such results. We state these results formally below.

**Corollary 3.** *Let $\mathbb{P}, \mathbb{Q}$ be two probability measures supported on $X \subseteq \mathbb{R}^d$, let $p$ and $q$ be the corresponding densities, and let $\mu_p$, $\mu_q$ and $\Sigma_p$, $\Sigma_q$ be the means and variances of $\mathbb{P}$ and $\mathbb{Q}$ respectively. Let $u_{\mathsf{F}} = \arg\max_u \mathsf{Fish}(\mathbb{P}, \mathbb{Q}; u)$, $b_{\mathsf{F}} = u_{\mathsf{F}}^\top \mu_p - \mathsf{MPM}(\mathbb{P}, \mathbb{Q}; u_{\mathsf{F}}) \sqrt{u_{\mathsf{F}}^\top \Sigma_p u_{\mathsf{F}}}$, $u_{\mathsf{M}} = \arg\max_u \mathsf{MPM}(\mathbb{P}, \mathbb{Q}; u)$; and $b_{\mathsf{M}} = u_{\mathsf{M}}^\top \mu_p - \mathsf{MPM}(\mathbb{P}, \mathbb{Q}; u_{\mathsf{M}}) \sqrt{u_{\mathsf{M}}^\top \Sigma_p u_{\mathsf{M}}}$. Suppose we have the linear classifiers $f_{\mathsf{F}}(X) = \text{sign}\left(u_{\mathsf{F}}^\top X + b_{\mathsf{F}}\right)$ and $f_{\mathsf{M}}(X) = \text{sign}\left(u_{\mathsf{M}}^\top X + b_{\mathsf{M}}\right)$ with accuracies $\alpha_{\mathsf{F}}$ and $\alpha_{\mathsf{M}}$ respectively. Then*

$$\alpha_{\mathsf{F}} \geq 1 - 2R^*(\mathbb{P}, \mathbb{Q}) \quad and \quad \alpha_{\mathsf{M}} \leq \frac{2(1 - 2R^*(\mathbb{P}, \mathbb{Q}))}{(1 - \sqrt{1 - 2R^*(\mathbb{P}, \mathbb{Q})})^2 + 2(1 - 2R^*(\mathbb{P}, \mathbb{Q}))}, \quad (8)$$

*where $R^*(\mathbb{P}, \mathbb{Q})$ denotes the Bayes error rate for the distributions $\mathbb{P}$ and $\mathbb{Q}$.*

The proof for this result is provided in Appendix B.

This perspective also allows us to develop a sanity check on the validity of our bounds. Suppose the Bayes classifier for a pair of distributions is $\text{sign}(g(X))$ for some $g \in \mathcal{F}$, where $\mathcal{F}$ is a family of witness functions. Then, optimizing the bound provided in Theorem 1 yields the Bayes classifier. For instance, if we choose $\mathbb{P} \equiv \mathcal{N}(\mu_p, \Sigma)$ and $\mathbb{Q} \equiv \mathcal{N}(\mu_q, \Sigma)$, then, maximizing the lower bounds obtained in Corollaries 1 and 2 yield the same solutions, which in turn maximizes the TV distance between $\mathbb{P}$ and $\mathbb{Q}$. We detail this in Appendix B.

## 7 EMPIRICAL EVALUATIONS

In this section, we demonstrate the utility of our lower bound, and the WITNESSPRUNEfamily of algorithms as a tool for structured pruning. Additional experimental details, including the experiment setup, are provided in D

### 7.1 MEASURING THE TV DISTANCE BETWEEN LINEARLY INSEPARABLE DATA

In this section, we use the bounds proposed in corollaries 1 and 2 to bound the TV distance between poorly separated datasets. We choose the 'Two Spirals' Dataset, and a dataset consisting of two zero-mean Gaussians with different variances. We choose $f(x) = u^\top (\varphi(X) - (\bar{\varphi}_1 - \bar{\varphi}_0)/2)$, where $\phi(X)$, when chosen to be of degree $d \geq 1$, is given by $\varphi(X) = 1 + (1^\top X) + \cdots + (1^\top X)^d$. We compare two methods, the lower bounds given in equation 5 and equation 6. We present our results in Figure 1. We observe that using the lower bound for Fisher (equation 5 outperforms the MPM lower bound (equation 6 on both toy datasets. However, the choice of polynomial witness functions clearly outperforms lower degree choices, such as using the Hellinger lower bound, particularly in the 'Two Gaussians' case.

### 7.2 VERIFYING CLASS-CONDITIONAL FEATURE MAPS ARE NOT GAUSSIAN

In this section, we attempt to validate the assumptions made in Murti et al. (2022) about the normality of the class-conditional feature distributions. To do so, we apply the Shapiro-Wilks test (Shapiro & Wilk, 1965), a standard test for Normality.

**Experiment Setup:** We consider a VGG16 model trained on CIFAR10. Let $\mathsf{BN}_j^l(X) = 1^\top Y_j^l(X)$. For each $l, j$, we collect 100 samples from each class $c \in [10]$. We then apply the Shapiro-Wilk

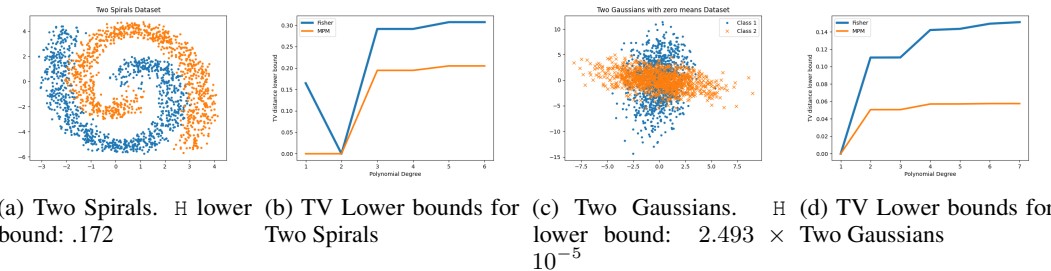

(a) Two Spirals. H lower bound: .172

(b) TV Lower bounds for Two Spirals

(c) Two Gaussians. H lower bound: $2.493 \times 10^{-5}$

(d) TV Lower bounds for Two Gaussians

Figure 1: Comparison of the performance of WITNESSPRUNE F with WITNESSPRUNE M with polynomial features on the TwoSpirals and Zero-Means Gaussians datasets.

normality test (Shapiro & Wilk, 1965), and we compute $p_{j,c}^l$ values, which are the minimum $p$-values from the Shapiro-Wilks test computed for the features of the $j$th filter in layer $l$ conditioned on class $c$. We consider that a filter's features are unlikely to be Gaussian if $p_{j,c}^l < 0.1$. We plot the heatmaps of $p_j^l = \min_{c \in [C]} p_{j,c}^l$ values for 15 randomly selected filters in Figure 2, to indicate the normality of the least Gaussian class-conditional features.

**Results:** We observe that for most layers, particularly those close to the output, the class-conditional feature distributions are highly unlikely to have been drawn from a Gaussian, with $p_{j,c}^l$ values for layers 10-12 in VGG16 typically being below $1e-5$. For layers with filters that yield likely-Gaussian features, we observe that for the majority of filters, at least one feature output is likely to be non-Gaussian. Moreover, the layers that generate Gaussian features are the hard-to-prune layers identified in (Murti et al., 2022; Liebenwein et al., 2019), which were shown in Murti et al. (2022) to be those features with poorly separated means.

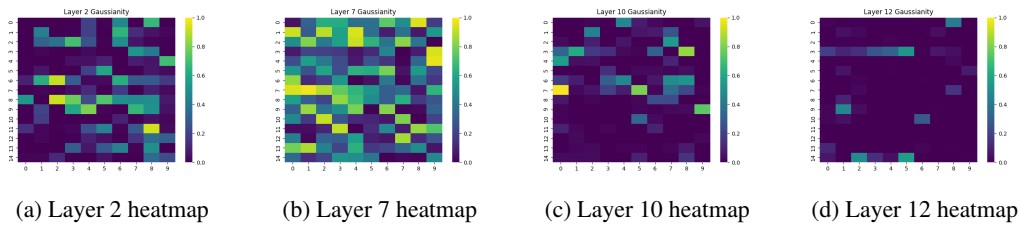

(a) Layer 2 heatmap

(b) Layer 7 heatmap

(c) Layer 10 heatmap

(d) Layer 12 heatmap

Figure 2: $p$-Value heatmaps from Shapiro-Wilks tests as applied to class-conditional features from different layers. $x$-axis is the class index, and $y$-axis is the filter index

### 7.3 EFFECTIVE PRUNING OF HARD-TO-PRUNE LAYERS

In this section, we utilize the lower bounds provided in this paper to prune hard-to-prune layers in neural networks. As noted in Murti et al. (2022); Liebenwein et al. (2019), some layers, in particular the initial layers in the case of VGG-nets, are difficult to effectively sparsify. In this set of experiments, we aim to show that using the lower bounds proposed in this work, we are able to better identify discriminative filters in hard-to-prune layers, and therefore prune those layers more effectively.

**Problem Setup:** We select a VGG16 model trained on CIFAR10. We fix pruning budgets of 40%, 50%, 60%, 70%, 80%. For each model, we then prune three hard-to-prune layers in isolation, and measure the impact on accuracy. We compare the following methods:

**TVSPrune:** We modify TVSPrune to prune a fixed budget, and using the BatchNorm random variables as described in Appendix C.
**WITNESSPRUNE-EQ:** We apply Algorithm 4 as presented in Appendix C using the features $\varphi(1^\top X) = [1^\top X, (1^\top X)^2]$. Thus, for each $c$, $f_{(c)}(X) = u^\top \left( \varphi(X) - (\bar{\varphi}_c + \varphi_{\bar{c}})/2) \right)$.
$L_1$-**based Pruning:** We use the $L_1$ norms of the filter weights, as proposed in Li et al. (2017).

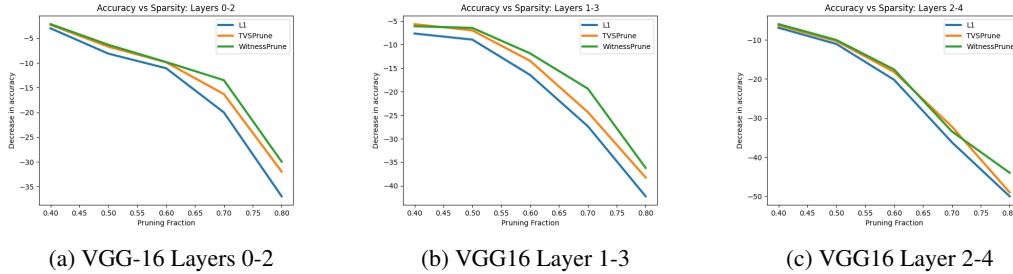

| (a) VGG-16 Layers 0-2 | (b) VGG16 Layer 1-3 | (c) VGG16 Layer 2-4 |

Figure 3: Comparison of the performance of WITNESSPRUNE F with TVSPrune and L1 pruning on hard-to-prune layers in VGG16 trained on CIFAR10

Table 1: WITNESSPRUNE  Performance with fine-tuning

| Model | Param. Sparsity | WITNESSPRUNE(our work) | TVSPRUNE(Murti et al., 2022) | CHIP (Sui et al., 2021) | L1 |
|---|---|---|---|---|---|
| VGG16 | 61.2% | **-0.37** % | -0.98% | -0.73% | 1.26% |
|  | 75.05% | **-1.32%** | -1.54% | -1.62 | -2.31 |
| VGG19 | 72.4% | **-.12%** | -.16% | N/A | -2.41% |
|  | 76.1% | **-.96%** | -1.13% | N/A | -3.30% |
| ResNet | 60.7% | **-1.21%** | -1.92% | -1.77% | -6.21% |

**Results and Discussion**   We present our results in Figure 3 The experiments show that WIT-NESSPRUNE  variants using quadratic features (using algorithms outperforms both TVSPrune and the $L_1$-norm based pruning strategy. In particular, we see that at $70\%$ sparsity in Layers 1-3, the models obtained by WITNESSPRUNE-EQ are $6.6\%$ more accurate than those obtained using TVSPrune.

## 7.4    EFFECTIVENESS OF THE WITNESSPRUNE ALGORITHM

In this section, we investigate the ability for WITNESSPRUNE  to sparsify models effectively without fine-tuning.  Specifically, we prune VGG16, VGG19, and ResNet56 models trained on CIFAR10 with two sets of fixed sparsity budgets, and then fine-tune them for 50 epochs. We show that models trained with  WITNESSPRUNE  are able to almost fully recover the accuracy of the original models. Table 1 shows the drop in accuracy for different sparsity levels after fine-tuning.

## 8    CONCLUSIONS

In this work, in Theorem 1, we propose new witness function-based lower bounds on the Total Variation distance between arbitrary distributions and apply them to the problem of distributional structured pruning. Our lower bounds are robust to the choice of distribution, and yield a family of pruning algorithms, WITNESSPRUNE( based on Algorithm 1), that require no access to the training set or loss function. Moreover, the bounds can be further generalized, as Corollaries 1 and 2 give bounds in terms of discriminant functions that can be kernelized.

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

## A   LOWER BOUNDS FOR TV AND HELLINGER DIVERGENCES

In this section, we derive lower bounds on other $f$-divergences using our witness function-based approach.

### A.1   LOWER BOUNDS ON THE TV DISTANCE

In this section, we provide the proof for a variant of Theorem 1.

**Theorem.** *Let $\mathbb{P}, \mathbb{Q}$ be two probability measures supported on $X \subseteq \mathbb{R}^d$, and let $p$ and $q$ be the corresponding densities. Let $\mathcal{F}$ be the set of functions with bounded first and second moments defined on $X$. Then,*

$$\mathrm{TV}(\mathbb{P}, \mathbb{Q}) \geq \sup_{f \in \mathcal{F}} \frac{\left(\bar{f}_p - \bar{f}_q\right)^2}{2\left(\bar{f}_p^{(2)} + \bar{f}_q^{(2)}\right)} \tag{9}$$

*Proof.* Choose an arbitrary $f \in \mathcal{F}$. Then, we have

$$
\begin{aligned}
\left(\bar{f}_p - \bar{f}_q\right)^2 &= \left(\int_{\mathcal{X}} (p(x) - q(x)) \, f(x) dx\right)^2 \\
&= \left(\int_{\mathcal{X}} \left(\sqrt{|p(x) - q(x)|}\sqrt{|p(x) - q(x)|}\right) f(x) dx\right)^2 \\
&\leq \left(\sqrt{\int_{\mathcal{X}} |p(x) - q(x)| dx}\right)^2 \left(\sqrt{\int_{\mathcal{X}} |p(x) - q(x)| f(x)^2 dx}\right)^2 \quad \text{(by Cauchy-Schwarz)} \\
&= 2\mathrm{TV}(\mathbb{P}, \mathbb{Q}) \left(\int_{\mathcal{X}} |p(x) - q(x)| f(x)^2 dx\right) \quad \text{(by Definition 2)} \\
&\leq 2\mathrm{TV}(\mathbb{P}, \mathbb{Q}) \left(\int_{\mathcal{X}} (p(x) + q(x)) \, f(x)^2 dx\right) \\
&= 2\mathrm{TV}(\mathbb{P}, \mathbb{Q}) \left(\bar{f}_p^{(2)} + \bar{f}_q^{(2)}\right).
\end{aligned}
$$

Thus, for any arbitrary $f \in \mathcal{F}$, we have

$$2\mathrm{TV}(\mathbb{P}, \mathbb{Q}) \geq \frac{\left(\bar{f}_p - \bar{f}_q\right)^2}{\left(\bar{f}_p^{(2)} + \bar{f}_q^{(2)}\right)},$$

from which it follows that

$$2\mathrm{TV}(\mathbb{P}, \mathbb{Q}) \geq \sup_{f \in \mathcal{F}} \frac{\left(\bar{f}_p - \bar{f}_q\right)^2}{\left(\bar{f}_p^{(2)} + \bar{f}_q^{(2)}\right)}. \tag{10}$$

$\square$

The proof of Theorem 1 follows from the fact that we can choose $f(x) = u^\top \varphi(x)$, and apply the formula for the expectation. Then, maximizing over $\mathcal{F}$ is equivalent to maximizing over $u$. Thus, the proof is completed.

## A.2 LOWER BOUNDS ON THE HELLINGER DISTANCE

In this section, we derive lower bounds on the Squared Hellinger distance based on the moments of witness functions.

First, we define the Hellinger distance as follows.

**Definition 3.** *Let* $\mathbb{P}, \mathbb{Q}$ *be two probability measures supported on* $\mathbb{R}^d$*, and let* $p$ *and* $q$ *be the corresponding densities. We define the squared Hellinger distance as*

$$\mathrm{H}^2(\mathbb{P}, \mathbb{Q}) = \frac{1}{2} \int_{\mathbb{R}^d} \left( \sqrt{p(x)} - \sqrt{q(x)} \right)^2 dx.$$

We state the lower bound formally below.

**Theorem.** *Let* $\mathbb{P}, \mathbb{Q}$ *be two probability measures supported on* $X \subseteq \mathbb{R}^d$*, and let* $p$ *and* $q$ *be the corresponding densities. Let* $\mathcal{F}$ *be the set of functions with bounded first and second moments defined on* $X$*. Then,*

$$2\mathrm{H}^2(\mathbb{P}, \mathbb{Q}) \geq \sup_{f \in \mathcal{F}} \frac{\left( \bar{f}_p - \bar{f}_q \right)^2}{\left( \sqrt{\bar{f}_p^{(2)}} + \sqrt{\bar{f}_q^{(2)}} \right)^2}. \tag{11}$$

*Proof.* Choose an arbitrary $f \in \mathcal{F}$. Then, we have

$$
\begin{aligned}
\left( \bar{f}_p - \bar{f}_q \right)^2 &= \left( \int_{\mathcal{X}} (p(x) - q(x)) f(x) dx \right)^2 \\
&= \left( \int_{\mathcal{X}} \left( \sqrt{p(x)} - \sqrt{q(x)} \right) \left( \sqrt{p(x)} + \sqrt{q(x)} \right) f(x) dx \right)^2 \\
&\leq \left( \int_{\mathcal{X}} \left( \sqrt{p(x)} - \sqrt{q(x)} \right)^2 dx \right) \left( \int_{\mathcal{X}} \left( \sqrt{p(x)} + \sqrt{q(x)} \right)^2 f(x)^2 dx \right) \quad \text{(by Cauchy-Schwarz)} \\
&= 2\mathrm{H}^2(\mathbb{P}, \mathbb{Q}) \left( \int_{\mathcal{X}} \left( p(x) + q(x) + 2\sqrt{p(x)q(x)} \right) f(x)^2 dx \right) \quad \text{(by Definition 3)} \\
&= 2\mathrm{H}^2(\mathbb{P}, \mathbb{Q}) \left( \bar{f}_p^{(2)} + \bar{f}_q^{(2)} + 2 \int_{\mathcal{X}} \sqrt{p(x)q(x)} f(x)^2 dx \right) \\
&\leq 2\mathrm{H}^2(\mathbb{P}, \mathbb{Q}) \left( \bar{f}_p^{(2)} + \bar{f}_q^{(2)} + 2 \sqrt{\int_{\mathcal{X}} p(x) f(x)^2 dx} \sqrt{\int_{\mathcal{X}} q(x) f(x)^2 dx} \right) \quad \text{(by Cauchy-Schwarz)} \\
&= 2\mathrm{H}^2(\mathbb{P}, \mathbb{Q}) \left( \bar{f}_p^{(2)} + \bar{f}_q^{(2)} + 2 \sqrt{\bar{f}_p^{(2)}} \sqrt{\bar{f}_q^{(2)}} \right) \\
&= 2\mathrm{H}^2(\mathbb{P}, \mathbb{Q}) \left( \sqrt{\bar{f}_p^{(2)}} + \sqrt{\bar{f}_q^{(2)}} \right)^2.
\end{aligned}
$$

Thus, for any arbitrary $f \in \mathcal{F}$, we have

$$2\mathrm{H}^2(\mathbb{P}, \mathbb{Q}) \geq \frac{\left( \bar{f}_p - \bar{f}_q \right)^2}{\left( \sqrt{\bar{f}_p^{(2)}} + \sqrt{\bar{f}_q^{(2)}} \right)^2},$$

from which it follows that

$$2\mathrm{H}^2(\mathbb{P}, \mathbb{Q}) \geq \sup_{f \in \mathcal{F}} \frac{\left( \bar{f}_p - \bar{f}_q \right)^2}{\left( \sqrt{\bar{f}_p^{(2)}} + \sqrt{\bar{f}_q^{(2)}} \right)^2}. \tag{12}$$

$\square$

# B    DISCRIMINANT FUNCTION BASED LOWER BOUNDS ON THE TV DISTANCE

In this section, we provide proofs for the discriminant-based lower bounds on the TV distance.

## B.1    FISHER DISCRIMINANT BASED LOWER BOUNDS

In this section, we provide proofs for Corollaries 1 and 3. We begin with the proof for Corollary 1.

**Corollary.** *Let $\mathbb{P}, \mathbb{Q}$ be two probability measures supported on $X \subseteq \mathbb{R}^d$, let $p$ and $q$ be the corresponding densities, and let $\mu_p$, $\mu_q$ and $\Sigma_p$, $\Sigma_q$ be the means and variances of $\mathbb{P}$ and $\mathbb{Q}$ respectively. Then,*

$$\mathrm{TV}(\mathbb{P}, \mathbb{Q}) \geq \sup_{u \in \mathbb{R}^d} \frac{\mathsf{Fish}(\mathbb{P}, \mathbb{Q}; u)}{2 + \mathsf{Fish}(\mathbb{P}, \mathbb{Q}; u)}. \tag{13}$$

*Proof.* Begin by choosing

$$f = u^\top \left( x - \frac{\mu_p + \mu_q}{2} \right),$$

where $u \in \mathbb{R}^d$ is constant. Then,

$$\bar{f}_p = \frac{1}{2} u^\top (\mu_p - \mu_2) \quad \text{and} \quad \bar{f}_q = \frac{1}{2} u^\top (\mu_q - \mu_p),$$

and

$$\begin{aligned}
\bar{f}_p^{(2)} &= u^\top \left( \mathbb{E}_{\mathbb{P}} \left[ \left( x - \frac{\mu_p + \mu_q}{2} \right) \left( x - \frac{\mu_p + \mu_q}{2} \right)^\top \right] \right) u \\
&= u^\top \left( \mathbb{E}_{\mathbb{P}} \left[ (x - \Delta)(x - \Delta)^\top \right] \right) u \quad (\text{setting } \Delta = \frac{\mu_p + \mu_q}{2}) \\
&= u^\top \left( \mathbb{E}_{\mathbb{P}} \left[ (x - \mu_p + \mu_p - \Delta)(x - \mu_p + \mu_p - \Delta)^\top \right] \right) u \\
&= u^\top \left( \Sigma_p + \frac{1}{4}(\mu_p - \mu_q)(\mu_p - \mu_q)^\top \right) u.
\end{aligned}$$

Similarly, we have

$$\bar{f}_q^{(2)} = u^\top \left( \Sigma_q + \frac{1}{4}(\mu_p - \mu_q)(\mu_p - \mu_q)^\top \right) u.$$

Substituting this into equation 9, we get

$$\begin{aligned}
2\mathrm{TV}(\mathbb{P}, \mathbb{Q}) &\geq \sup_{u \in \mathbb{R}^d} \frac{\left( u^\top (\mu_p - \mu_q) \right)^2}{u^\top (\Sigma_p + \Sigma_q) u + \frac{1}{2} \left( u^\top (\mu_p - \mu_q) \right)^2} \\
&\geq \sup_{u \in \mathbb{R}^d} \frac{2 \left( u^\top (\mu_p - \mu_q) \right)^2}{2 u^\top (\Sigma_p + \Sigma_q) u + \left( u^\top (\mu_p - \mu_q) \right)^2} \\
&\geq \sup_{u \in \mathbb{R}^d} \frac{2 \frac{\left( u^\top (\mu_p - \mu_q) \right)^2}{u^\top (\Sigma_p + \Sigma_q) u}}{2 + \frac{\left( u^\top (\mu_p - \mu_q) \right)^2}{u^\top (\Sigma_p + \Sigma_q) u}} \\
\Rightarrow \mathrm{TV}(\mathbb{P}, \mathbb{Q}) &\geq \sup_{u \in \mathbb{R}^d} \frac{\mathsf{Fish}(\mathbb{P}, \mathbb{Q}; u)}{2 + \mathsf{Fish}(\mathbb{P}, \mathbb{Q}; u)}
\end{aligned}$$

$\square$

We next provide the proof for Corollary 3.

**Corollary.** *Let $\mathbb{P}, \mathbb{Q}$ be two probability measures supported on $X \subseteq \mathbb{R}^d$, let $p$ and $q$ be the corresponding densities, and let $\mu_p$, $\mu_q$ and $\Sigma_p$, $\Sigma_q$ be the means and variances of $\mathbb{P}$ and $\mathbb{Q}$ respectively. Let $\mathsf{Fish}^*(\mathbb{P}, \mathbb{Q}) = \arg\max_u \mathsf{Fish}(\mathbb{P}, \mathbb{Q}; u)$. Then, the Bayes Error $R^*(\mathbb{P}, \mathbb{Q})$ satisfies*

$$R^*(\mathbb{P}, \mathbb{Q}) \leq \frac{1}{\mathsf{Fish}^*(\mathbb{P}, \mathbb{Q}) + 2}.$$

*Furthermore, suppose we have a linear classifier $f(x) = u_\dagger^\top x + b_\dagger$, where $u_\dagger = \arg\min_u$ $\mathsf{Fish}(\mathbb{P}, \mathbb{Q}; u)$ and $b_\dagger = u_\dagger^\top \mu_p - \mathsf{MPM}(\mathbb{P}, \mathbb{Q}; u_\dagger)\sqrt{u_\dagger^\top \Sigma_p u_\dagger}$. Then, the accuracy of this classifier $\alpha_\dagger$ satisfies*

$$\alpha_\dagger \geq 1 - 2R^*(\mathbb{P}, \mathbb{Q})$$

*Proof.* Recall that the Bayes risk is related to the TV distance as

$$R^*(\mathbb{P}, \mathbb{Q}) = \frac{1}{2}\left(1 - \mathrm{TV}(\mathbb{P}, \mathbb{Q})\right).$$

Substituting the expression from Corollary 1, we get

$$R^*(\mathbb{P}, \mathbb{Q}) \leq \frac{1}{2}\left(1 - \frac{\mathsf{Fish}^*(\mathbb{P}, \mathbb{Q})}{\mathsf{Fish}^*(\mathbb{P}, \mathbb{Q}) + 2}\right) = \frac{1}{\mathsf{Fish}^*(\mathbb{P}, \mathbb{Q}) + 2}$$

Next, from Lanckriet et al. (2002), we have

$$1 - \alpha_\dagger \leq \frac{2}{2 + \mathsf{Fish}^*(\mathbb{P}, \mathbb{Q})} = 2R^*(\mathbb{P}, \mathbb{Q}).$$

Rearranging the terms, we prove the statement. $\qquad\square$

### B.2 Minimax Probability Machine Based Lower Bounds

In this section, we provide proofs for Corollaries 2 and the second statement in 3. We begin with the proof for Corollary 2.

**Corollary.** *Let $\mathbb{P}, \mathbb{Q}$ be two probability measures supported on $X \subseteq \mathbb{R}^d$, let $p$ and $q$ be the corresponding densities, and let $\mu_p$, $\mu_q$ and $\Sigma_p$, $\Sigma_q$ be the means and variances of $\mathbb{P}$ and $\mathbb{Q}$ respectively. Then,*

$$\sqrt{\mathrm{TV}(\mathbb{P}, \mathbb{Q})} \geq \sup_{u \in \mathbb{R}^d} \frac{\mathsf{MPM}(\mathbb{P}, \mathbb{Q}; u)}{\sqrt{2} + \mathsf{MPM}(\mathbb{P}, \mathbb{Q}; u)}$$

*Proof.* As with Corollary 1, choose

$$f = u^\top \left(x - \frac{\mu_p + \mu_q}{2}\right),$$

where $u \in \mathbb{R}^d$ is constant. Then,

$$\bar{f}_p = \frac{1}{2}u^\top(\mu_p - \mu_2) \quad \text{and} \quad \bar{f}_q = \frac{1}{2}u^\top(\mu_q - \mu_p).$$

Then, we get

$$\bar{f}_p^{(2)} = u^\top \Sigma_p u + \frac{1}{4}\left(u^\top(\mu_p - \mu_q)\right)^2 \quad \text{and} \bar{f}_q^{(2)} = u^\top \Sigma_q u + \frac{1}{4}\left(u^\top(\mu_p - \mu_q)\right)^2.$$

Plugging these expressions into equation 9, we get

$$2\mathrm{TV}(\mathbb{P}, \mathbb{Q}) \geq \sup_{u \in \mathbb{R}^d} \frac{\left(u^\top(\mu_p - \mu_q)\right)^2}{u^\top(\Sigma_p + \Sigma_q)u + \frac{1}{2}\left(u^\top(\mu_p - \mu_q)\right)^2}$$

$$\geq \sup_{u \in \mathbb{R}^d} \frac{2\left(u^\top(\mu_p - \mu_q)\right)^2}{2u^\top(\Sigma_p + \Sigma_q)u + \left(u^\top(\mu_p - \mu_q)\right)^2}$$

$$\Rightarrow \sqrt{\mathrm{TV}(\mathbb{P}, \mathbb{Q})} \geq \sup_{u \in \mathbb{R}^d} \frac{|u^\top(\mu_p - \mu_q)|}{\sqrt{2u^\top(\Sigma_p + \Sigma_q)u + \left(u^\top(\mu_p - \mu_q)\right)^2}}$$

$$\geq \sup_{u \in \mathbb{R}^d} \frac{|u^\top(\mu_p - \mu_q)|}{\sqrt{2}\left(\sqrt{u^\top \Sigma_p u} + \sqrt{u^\top \Sigma_q u}\right) + |u^\top(\mu_p - \mu_q)|}$$

$$\geq \sup_{u \in \mathbb{R}^d} \frac{\frac{|u^\top(\mu_p - \mu_q)|}{\sqrt{u^\top \Sigma_p u} + \sqrt{u^\top \Sigma_q u}}}{\sqrt{2} + \frac{|u^\top(\mu_p - \mu_q)|}{\sqrt{u^\top \Sigma_p u} + \sqrt{u^\top \Sigma_q u}}} = \sup_{u \in \mathbb{R}^d} \frac{\mathsf{MPM}(\mathbb{P}, \mathbb{Q}; u)}{\sqrt{2} + \mathsf{MPM}(\mathbb{P}, \mathbb{Q}; u)}$$

$\qquad\square$

We next present the proof for Corollary 3.

**Corollary.** *Let $\mathbb{P}, \mathbb{Q}$ be two probability measures supported on $X \subseteq \mathbb{R}^d$, let $p$ and $q$ be the corresponding densities, and let $\mu_p$, $\mu_q$ and $\Sigma_p$, $\Sigma_q$ be the means and variances of $\mathbb{P}$ and $\mathbb{Q}$ respectively. Suppose $u_* = \arg\max_u \mathsf{MPM}(\mathbb{P}, \mathbb{Q}; u)$ and $b_* = u_*^\top \mu_p - \mathsf{MPM}(\mathbb{P}, \mathbb{Q}; u_*)\sqrt{u_*^\top \Sigma_p u_*}$. Then, the generalization error $\alpha_*$ of a classifier $f^*(x) = u_*^\top x + b_*$ satisfies*

$$\alpha_* \geq \frac{2(1 - 2R^*(\mathbb{P}, \mathbb{Q}))}{(1 - \sqrt{1 - 2R^*(\mathbb{P}, \mathbb{Q})})^2 + 2(1 - 2R^*(\mathbb{P}, \mathbb{Q}) - 1)},$$

*and*

$$R^*(\mathbb{P}, \mathbb{Q}) \leq \frac{2 - 2\alpha_* + 2\sqrt{2\alpha_* - 2\alpha_*^2}}{2 - \alpha_* + 2\sqrt{2\alpha_* - 2\alpha_*^2}}. \tag{14}$$

*Proof.* The proof follows by rearranging the terms of equation 6 to get an expression for $\sqrt{\mathsf{TV}(\cdot)}$, and recalling that

$$\mathsf{MPM}(\mathbb{P}, \mathbb{Q}; u_*) = \sqrt{\frac{\alpha_*}{1 - \alpha_*}}.$$

Then, we obtain an expression for $\alpha_*$ in terms of the TV distance, and reverse the inequality with the expression $R^*(\mathbb{P}, \mathbb{Q}) = \frac{1}{2}(1 - \mathsf{TV}(\mathbb{P}, \mathbb{Q}))$. Rearrange the terms to complete the proof. □

### B.3   Computing $\mathsf{TV}(\mathbb{P}, \mathbb{Q})$ from the Lower Bound

The lower bound proposed in Theorem 1 is not tight, as the Cauchy-Schwarz inequality used in the derivation of the bound is only not strict when the witness function $f$ is a constant. However, there are cases where the bound can be used to compute the true TV distance. We state one such case below in Corollary 4.

**Corollary 4.** *Suppose Suppose $\mathbb{P} \equiv \mathcal{N}(\mu_p, \Sigma)$ and $\mathbb{Q} \equiv \mathcal{N}(\mu_q, \Sigma)$ Let $f(x; u) = u^\top(x - \frac{1}{2}(\mu_p - \mu_q))$ be a witness function. Then,*

$$\mathsf{TV}(\mathbb{P}, \mathbb{Q}) = 2\Phi\left(\sqrt{(u^*)^\top(\mu_p - \mu_q)}/2\right) - 1,$$

*where*

$$u^* = \arg\max_u \frac{\left(\mathbb{E}_{x \sim \mathbb{P}}[f(x; u)] - \mathbb{E}_{x \sim \mathbb{Q}}[f(x; u)]\right)^2}{\mathbb{E}_{x \sim \mathbb{P}}[f(x; u)^2] + \mathbb{E}_{x \sim \mathbb{Q}}[f(x; u)^2]}$$

*Proof.* First, following the proof of Corollary 1, we have $u^* = \Sigma^{-1}(\mu_p - \mu_q)$. Substituting $u*$ into the expression $\mathsf{TV}(\mathbb{P}, \mathbb{Q}) = 2\Phi\left(\sqrt{(u^*)^\top(\mu_p - \mu_q)}/2\right) - 1$, we get $\mathsf{TV}(\mathbb{P}, \mathbb{Q}) = 2\Phi\left(\sqrt{(\mu_p - \mu_q)^\top \Sigma^{-1}(\mu_p - \mu_q)}/2\right) - 1$. Note that with this choice of $u*$, the square root term remains well-defined. This matches the well-known result for the TV distance between Gaussian measures with the same variance (Pardo, 2018). Thus, we prove the statement. □

*Remark:* This result also illustrates the case where the Bayes' classifier lies in the set of functions $\mathcal{F} := \{f(x) : f(x) = u^\top \varphi(x)\}$ for a given function $\varphi(x)$. In this case, if $\varphi(x) = x - \frac{1}{2}(\mu_p - \mu_q)$, and $\mathbb{P}$ and $\mathbb{Q}$ are Gaussian with the same variant, the Bayes classifier is equivalent to the Fisher discriminant.

*Remark:* This result and the observations one may draw from it also motivate another perspective on identifying discriminative filters without access to the training set or loss function. Specifically, we say that the discriminative ability of a filter can be measured as the classification accuracy of the best possible classifier - the Bayes classifier - of a model trained on the feature map generated by the filter. Since we cannot measure the TV distance (and thus, the Bayes error) directly, we may use our lower bound instead.

# C VARIANTS OF THE WITNESSPRUNE ALGORITHM

## C.1 FISHER-BASED LOWER BOUNDS AND TVSPRUNE

We choose $f(X) = u^\top \varphi(X)$. Let

$$\mu_{j,c}^l = \mathbb{E}_{X \sim \mathcal{D}_c} [\varphi(X)]$$

and let

$$\Sigma_{j,c}^l = \mathbb{E}_{X \sim \mathcal{D}_c} \left[ (\varphi(X) - \mu_{j,c}^l)(\varphi(X) - \mu_{j,c}^l)^\top \right].$$

Then, we get
If $\varphi(X)$ is a vector of quadratic functions of $X$, we call the algorithm WITNESSPRUNE-FQ.

---

**Algorithm 2:** WITNESSPRUNE-FQ

---

**Input:** Class conditional distributions $\mathcal{D}_c$, $c \in [C]$, Pretrained CNN with parameters
    $\mathcal{W} = (W_1, \cdots, W_L)$, layerwise sparsity budgets $B^l$, witness function $f$

**for** $l \in [L]$ **do**
    Set $S^l = [s_1^l, \cdots, s_{N_l}^l] = \mathbf{0}_{N_l}$
    Compute $\mu_{j,c}^l \mu_{j,\bar{c}}^l, \Sigma_{j,c}^l, \Sigma_{j,\bar{c}}^l$ for all $j, c$
    Compute

$$r_j^l = \max_u \frac{(u^\top (\mu_{j,c}^l - \mu_{j,\bar{c}}^l)^2)}{2u^\top (\Sigma_{j,c}^l + \Sigma_{j,\bar{c}}^l)^2)u}$$

    equation 7 for all $j$.
    **if** $j \in \text{sort}_{B_l}(\{r_j^l\}_{j=1}^{N_l})$ **then**
        Set $s_j^l = 1$

**Output:** Sparse masks $S^1, \cdots, S^L$
**return** $\hat{\mathcal{W}}$

---

## C.2 RECOVERING TVSPRUNE

In TVSPrune, at any layer $l$, the $j$th filter is pruned if

$$1 - e^{-\Delta_{l,j}}$$

where $\Delta_{l,j}$ is the minimum Fisher discriminant between pairs of classes. Suppose that we identify important and discriminative filters by measuring the TV distance in a pairwise sense. Recall that Corollary 1 gives us a bound that is also monotonic in $\Delta_{l,j}$. If we apply the strategy that we prune all filters with a score less than a threshold, we would prune filter $j$ if

$$\frac{\Delta}{2 + \Delta} \leq \gamma$$

for some $\gamma \in (0, 1)$. We can now find a relation between $\gamma$ and $\eta$. First, note that if $1 - e^{\frac{-\Delta}{4}} \leq \eta$, then $\Delta \leq 4(1 - \eta)$. Similarly, if we prune $\frac{\Delta}{2+\Delta} \leq \gamma$, then $\Delta \leq \frac{2\gamma}{1-\gamma}$. Equating the two gives us the expression

$$\eta = \frac{3\gamma - 2}{2\gamma - 2}.$$

Thus, both TVSPrune and a variant of WITNESSPRUNE-F where the TV distance is measured pairwise, and which prunes at a threshold are equivalent, as they require pruning the $j$th filter if

$$\Delta \leq 4 - 4\eta = \frac{3\gamma - 2}{2\gamma - 2}.$$

## C.3 USING THE BATCHNORM RANDOM VARIABLES

The BatchNorm random variables for this layer are given by

$$\mathsf{BN}^l(X) = \left[ \mathbf{1}^\top Y_1^l(X), \cdots, \mathbf{1}^\top Y_{N_l}^l(X) \right] = \left[ \mathsf{BN}_1^l(X), \cdots, \mathsf{BN}_{N_l}^l(X) \right]. \tag{15}$$

As stated earlier, our goal is to minimize the TV distance between the distributions of the pruned and unpruned features; we use the BatchNorm random variables as a proxy for the features $Y^l(X)$. Next, the moments of $\mathsf{BN}^l(X)$ are given by

$$\mathbb{E}_{X \sim \mathcal{D}}\left[\mathsf{BN}_i^l(X)\right] = \mathsf{BN}_i^l = \left[\mu_i^l\right] \quad \text{and} \quad \mathrm{Var}(\mathsf{BN}_i^l(X)) = (\sigma_i^l)^2. \tag{16}$$

Suppose $\mathsf{BN}^l(X)$ is drawn from the distribution $\mathcal{D}_j^{\mathsf{BN},l}$, $\mathcal{D}_{j,c}^{\mathsf{BN},l}$ be the $c$th class conditional distribution, and let $\mathcal{D}_{j,\bar{c}}^{\mathsf{BN},l}$ be the distribution of features sampled from the complement of class $c$.

## C.4 Minimax Probability Machine based Algorithms

We define $\mu_{j,c}^l \; \mu_{j,\bar{c}}^l, \Sigma_{j,c}^l, \Sigma_{j,\bar{c}}^l$ as previously. We then state the algorithm as follows.

---

**Algorithm 3:** WITNESSPRUNE-M

---

**Input:** Class conditional distributions $\mathcal{D}_c$, $c \in [C]$, Pretrained CNN with parameters
$\quad\quad \mathcal{W} = (W_1, \cdots, W_L)$, layerwise sparsity budgets $B^l$, witness function $f$

**for** $l \in [L]$ **do**
$\quad$ Set $S^l = [s_1^l, \cdots, s_{N_l}^l] = \mathbf{0}_{N_l}$
$\quad$ Compute $\mu_{j,c}^l \; \mu_{j,\bar{c}}^l, \Sigma_{j,c}^l, \Sigma_{j,\bar{c}}^l$ for all $j, c$
$\quad$ Compute

$$r_j^l = \max_u \frac{|u^\top(\mu_{j,c}^l - \mu_{j,\bar{c}}^l)|}{\sqrt{u^\top \Sigma_{j,c}^l u} + \sqrt{u^\top \Sigma_{j,\bar{c}}^l u}}$$

$\quad$ for all $j$.
$\quad$ **if** $j \in \mathrm{sort}_{B_l}(\{r_j^l\}_{j=1}^{N_l})$ **then**
$\quad\quad$ Set $s_j^l = 1$
**Output:** Sparse Masks $S_1, \cdots, S^L$
**return** $\hat{\mathcal{W}}$

---

If $\varphi(X)$ is a vector of quadratic functions of $X$, we call the algorithm WITNESSPRUNE-MQ.

## C.5 ENSEMBLEPRUNE - Taking the best of FISHERPRUNE and MPMPRUNE

We choose the same witness functions as we did for WITNESSPRUNE-M and WITNESSPRUNE-F, and define the moments in the same fashion. We then

---

**Algorithm 4:** WITNESSPRUNE-E

---

**Input:** Class conditional distributions $\mathcal{D}_c$, $c \in [C]$, Pretrained CNN with parameters
$\quad\quad \mathcal{W} = (W_1, \cdots, W_L)$, layerwise sparsity budgets $B^l$, witness function $f$

**for** $l \in [L]$ **do**
$\quad$ Set $S^l = [s_1^l, \cdots, s_{N_l}^l] = \mathbf{0}_{N_l}$
$\quad$ Compute $\mu_{j,c}^l \; \mu_{j,\bar{c}}^l, \Sigma_{j,c}^l, \Sigma_{j,\bar{c}}^l$ for all $j, c$
$\quad$ Compute

$$r_j^l = \max \left\{ \max_u \frac{|u^\top(\mu_{j,c}^l - \mu_{j,\bar{c}}^l)|}{\sqrt{u^\top \Sigma_{j,c}^l u} + \sqrt{u^\top \Sigma_{j,\bar{c}}^l u}}, \; \max_u \frac{(u^\top(\mu_{j,c}^l - \mu_{j,\bar{c}}^l)^2)}{2u^\top(\Sigma_{j,c}^l + \Sigma_{j,\bar{c}}^l)^2)u} \right\}$$

$\quad$ for all $j$.
$\quad$ **if** $j \in \mathrm{sort}_{B_l}(\{r_j^l\}_{j=1}^{N_l})$ **then**
$\quad\quad$ Set $s_j^l = 1$
**Output:** Sparse Masks $S_1, \cdots, S^L$
**return** $\hat{\mathcal{W}}$

---

# D    ADDITIONAL EXPERIMENTAL DETAILS

In this section, we detail additional experiments not mentioned in the main paper, as well as a comprehensive description of our experimental setup.

## D.1    PRUNING SETUP

In this section, we discuss our experimental setup.

### D.1.1    PLATFORM DETAILS

The hardware used for the experiments in this work are detailed below:

1. Server computer with 2 NVIDIA RTX3090Ti GPUs with Intel i9-12700 processors, running Ubuntu 20.04, with Python 3.11 and CUDA Tools 10.2 with PyTorch 2.0.1.
2. Desktop computer with 1 NVIDIA RTX3070Ti GPUs with Intel i7-10700 processor, running Ubuntu 22.04, with Python 3.11 and CUDA Tools 11.7 with PyTorch 2.0.1.

### D.1.2    MODELS UNDER CONSIDERATION

We consider the following models.

- **VGG16/19 trained on CIFAR10 and CIFAR100:** We use the pre-trained VGG11/16/19 models trained on CIFAR10 and CIFAR100. The models achieve accuracies greater than 90% on both datasets.
- **ResNet56 trained on CIFAR10:** We consider a ResNet56 model trained on CIFAR10. We do not prune layers that are part of complex interconnections (such as the final layer in each BasicBlock).

All our models were obtained from:
`https://github.com/chenyaofo/pytorch-cifar-models`

### D.1.3    DATASET SELECTION

Since we assume that the training dataset is unavailable to us, we utilize the validation set as a proxy for the data-distribution. We detail our dataset splits in Table 2.

Table 2: Breakdown of dataset splits used in our experiments.

| Dataset | Training Set | TV Distance Set | Test Set |
|---|---|---|---|
| CIFAR10 | Not used | 4000 images from test set | 6000 images from Test set |

### D.1.4    HYPERPARAMETER DETAILS

We detail the hyperparameters used in our experiments below.

1. **Batch Size:** 128
2. **Epochs:** 50
3. **Learning Rate:** .001
4. **Optimizer:** ADAM
5. **Weight Decay:** .0005
6. **Momentum paramters:** .9 and .99.

