# OpenReview forum: "Distributional Structured Pruning by Lower bounding the Total Variation Distance using Witness functions"
_ICLR.cc/2024/Conference — Submitted to ICLR 2024_

### Official Review · Reviewer_G7wT · 2023-10-29

**Soundness:** 3 good
**Presentation:** 3 good
**Contribution:** 2 fair
**Rating:** 6
**Confidence:** 3

**Summary:**

This paper introduces an innovative concept of 'witness functions,' providing a more accurate and reliable framework for estimating the Total Variation (TV) distance between probability distributions. This theoretical insight has practical implications, particularly in the realm of neural network pruning. By applying the witness function methodology, the authors develop 'WitnessPrune' which is an algorithm that efficiently identifies and eliminates redundant elements in neural networks, thereby optimizing their performance without sacrificing accuracy. The paper also bridges theoretical understanding and practical application by establishing insightful connections between discriminant-based classifiers and the TV distance, enhancing the robustness of machine learning models.

**Strengths:**

1. The paper introduces a pioneering approach to neural network pruning that deviates from the standard assumption of Gaussian-distributed class-conditional features. By employing a witness function-based strategy for lower bounding the Total Variation (TV) distance, the research not only addresses the limitations of previous methodologies but also enhances the precision of distributional pruning.
2. The introduction of the WITNESSPRUNE algorithms, derived from these theoretical foundations, which could optimize neural network performance significantly, particularly in resource-sensitive applications.

**Weaknesses:**

1.  The paper seems to lack extensive testing across a diverse range of neural network architectures and data distributions. This limitation raises concerns about the applicability of the proposed methods in various real-world scenarios and different neural network models.
2. The paper's methodologies rely on certain assumptions regarding witness functions and distribution moments, necessary for theoretical formulations but potentially unrealistic in practice.
3. VGG-16 is a very old NN architecture. It would be better if the authors could demonstrate that their approach works for newer NN architecture.

**Questions:**

1. How do these methods scale with increasingly complex and deep neural networks? Is there an in-depth analysis of the computational overhead and efficiency trade-offs when applying the proposed pruning techniques to larger-scale networks, especially compared to traditional pruning methods?
2. Could the authors provide further evidence on how the proposed pruning strategies perform across a wider range of scenarios? For instance, how do these methods fare with different types of neural network architectures, and varying data modalities?

---

> ### Author Response · Authors · 2023-11-17
> **Response to Reviewer G7wT**
>
> We thank the reviewer for the comments and discussion. Specifically:
>
> **The paper introduces a pioneering approach to neural network pruning that deviates from the standard assumption of Gaussian-distributed class-conditional features. By employing a witness function-based strategy for lower bounding the Total Variation (TV) distance, the research not only addresses the limitations of previous methodologies but also enhances the precision of distributional pruning.**
>
> We thank the author for their strong appreciation for our work!
>
> **The introduction of the WITNESSPRUNE algorithms, derived from these theoretical foundations, which could optimize neural network performance significantly, particularly in resource-sensitive applications.**
>
> We appreciate the attention to our well-founded pruning algorithms.
>
> **The paper seems to lack extensive testing across a diverse range of neural network architectures and data distributions. This limitation raises concerns about the applicability of the proposed methods in various real-world scenarios and different neural network models.**
>
> We will provide a new battery of experiments highlighting the efficacy of our algorithm to other datasets and architectures within the next couple of days. Specifically, we will present results evaluating our algorithm on ResNet50 models trained on ImageNet. We'll post those results here, and please check out the general response as well!
>
> **The paper's methodologies rely on certain assumptions regarding witness functions and distribution moments, necessary for theoretical formulations but potentially unrealistic in practice.**
>
> The assumptions made on (a) the distributions, and (b) the witness functions themselves, are very minimal. In Theorems 1 and 2, we only assume we have access to $k$ moments of  the distribution. Similarly, the witness functions themselves only need to have bounded first and second order moments; that is $\mathbb{E}[u^\top \varphi(X)] <\infty$ and $\mathbb{E}[(u^\top \varphi(X))^2] < \infty$.
>
> **VGG-16 is a very old NN architecture. It would be better if the authors could demonstrate that their approach works for newer NN architecture.**
>
> Further experiments are being conducted, and are coming soon! In particular, we compare the performance of WitnessPrune with CHIP [2] and TVSPrune [1] for pruning ResNet50 on the ImageNet dataset. We'll post those results both here and in the general response.
>
> **How do these methods scale with increasingly complex and deep neural networks? Is there an in-depth analysis of the computational overhead and efficiency trade-offs when applying the proposed pruning techniques to larger-scale networks, especially compared to traditional pruning methods?**
>
> We thank the reviewer for this insightful question. We answer the question in 2 parts.
> 1. Traditional structured pruning strategies that require gradients often require expensive backpropagation in order to compute saliency scores (which are used to identify which filters are important, and which are not). **In this work, we focus on the challenging use case where the training data and the loss function are unavailable to us**. As a benefit to this, we no longer need to compute the gradients.
> 2. The key computational cost involved in this algorithm is the storage of moments required to compute the $r^l_j$ scoress defined in Equation 7. Unlike previous works such as TVSPrune [1] and CHIP [2], we do not need to store the moments of entire feature maps - we need only store the first two moments of the witness function $f(X) = u^\top \varphi(X)$. Thus, prior to computing the $r^l_j$ scores, we only  need to store $2n_l$ real numbers per class for each layer. This is in contrast to [1] and [2], where moments of the entire (flattened) feature maps are stored. Furthermore, unlike prior work such as TVSPrune, for classification tasks where the dataset has $K$ classes, we need only store $2K$ moments, whereas TVSPrune requires the storage of ${K \choose 2}$ moments.
>
> **Could the authors provide further evidence on how the proposed pruning strategies perform across a wider range of scenarios? For instance, how do these methods fare with different types of neural network architectures, and varying data modalities?**
>
> We'll post additional experimental results both here and in the general response, in the next 48 hours. Our experiments will focus on ResNet50 model trained on the ImageNet dataset, and compare with some standard baselines [1],[2],[3].
>
> **References**
>
> [1] *TVSPrune - Pruning Non-discriminative filters via Total Variation separability of intermediate representations without fine tuning*. Murti et al, 2023.
>
> [2] *CHIP: CHannel Independence-based Pruning for Compact Neural Networks*. Sui et al, 2021.
>
> [3] *Pruning Filters for Efficient Convnets*. Li et al, 2017.

---

> ### Author Response · Authors · 2023-11-21
> **Response to Reviewer G7wT with Additional Experiments**
>
> ### **ResNet50 experiments on Imagenet**
>
> As requested by the reviewer, we have validated the proposed WitnessPrune algorithm on the ResNet50 model trained on ImageNet.
>
> We compare WitnessPrune with CHIP [1], TVSPrune (the variant discussed in Appendix C.3), and L1 [2]. In our experimental setup, we fine-tune our models for 120 epochs with a batch size of 128 (other hyperparameters to be uploaded in the amended draft, in Appendix D). We present our results below.
>
> | **Algorithm** | WitnessPrune | CHIP | **TVSPrune** | **L1** |
> |---|---|---|---|---|
> | **20.2% sparsity** (No FT) |  **-2.67%**|  -2.99%|  -2.85%| -6.45%|
> | **20.2% sparsity** (FT) |  **+.02%**	|  **+.01%**	|  -.01%	| -1.08% |
> | **40.2% sparsity** |  **-2.58%**	|  -2.74%	|  -2.77%	| -4.45% |
>
> **Observations**
>
> Our experiments indicate that WitnessPrune clearly outperforms several contemporary baselines on the ImageNet task on the ResNet50 architecture, demonstrating its broad applicability to classification tasks.
>
>
> **We hope we've addressed the concerns you raised. If you have additional questions, concerns, or thoughts, we are eager to engage in further discussions.**
>
> ### References
>
> [1] *TVSPrune - Pruning Non-discriminative filters via Total Variation separability of intermediate representations without fine tuning*. Murti et al, 2023.
>
> [2] *CHIP: CHannel Independence-based Pruning for Compact Neural Networks*. Sui et al, 2021.
>
> [3] *Pruning Filters for Efficient Convnets*. Li et al, 2017.

---

> > ### Comment · Reviewer_G7wT · 2023-11-21
> >
> > Thank you for replying to my concerns and conducting additional experiments. After checking other reviewers' comments and considering that I already gave a positive score, I would like to keep the current score.

---

### Official Review · Reviewer_XVjC · 2023-10-30

**Soundness:** 3 good
**Presentation:** 3 good
**Contribution:** 3 good
**Rating:** 6
**Confidence:** 3

**Summary:**

The paper discusses distributional structured pruning in deep neural networks by introducing new lower bounds on TV Distance that depends on moments of witness functions. The paper introduces a new algorithm "WitnessPrune" that utilizes the lower bounds for pruning. The paper discusses both theoretical justification of this algorithm and provides some experimental evidence supporting the effectiveness of WitnessPrune in pruning.

**Strengths:**

Originality and significance: While the topic of pruning has existed within the literature, the paper presents interesting contributions, both theoretically and empirically, that were shown to improve upon state of the art. Judging from this, the paper presents good originality and significance, and the topic is relevant to ICLR.

Quality and clarity: Overall, the quality and clarity of the paper is decent, but can be improved with further revision. The paper effectively guides the reader via usage of questions and discussions. There are some issues surrounding the formatting of references, typos, etc., though. More suggestions regarding quality and clarity are presented in the following section.

Interesting-ness of results: The proofs and techniques presented within the paper is also interesting and insightful by itself. For example, the proofs stated in sections A.1 and A.2 in the appendix are simple, clean and to the point.

**Weaknesses:**

Overall, there are not too many major weaknesses. Some additional questions regarding technicalities are listed in the following "Questions" section. Here, only weaknesses in overall presentation are discussed.

Potential breach of anonymity: In Table 1, there is a column with header "TVSPrune(Murti et al., 2022) (our work)".

Clarity issues: In certain parts of the paper, clarity issues cause confusion when reading. Some examples include:
- Formatting of references. The paper chooses to use references without separations from text. While this is suitable for some scenarios, in other situations it obstructs the flow of the reading, and it may be beneficial to update the references in these situations. This can likely be changed quite easily.
- Unclear definitions. There are multiple sections of the paper that can be revised to make definitions clearer. One example is in Section 2.2: the very first sentence abruptly ends with a "..., and let."
Formatting issues and typos throughout the paper should be revised and updated.

**Questions:**

Some minor questions for clarification and discussions:

1. In Section 4.1, there is an assumption that that functions "g" and "G" exist. How strict is this assumption and how does it affect the main results?

2. Are there theoretical justifications on whether WitnessPrune is generally preferable over previous methods such as TVSPrune?

3. Regarding Table 1, why is CHIP unavailable for VGG19? Is it possible to produce the numbers for that experiment? Also, regarding Table 1, how are the sparsity levels in the "Param. Sparsity" selected?

4. While there are some discussions on the variants of WitnessPrune in the appendix, it is not discussed thoroughly and the clarity there is lacking. Are there additional experiments to discuss the effectiveness of these variants?

**Details Of Ethics Concerns:**

There is a potential breach of anonymity: in Table 1, there is a column with header "TVSPrune(Murti et al., 2022) (our work)".

---

> ### Author Response · Authors · 2023-11-17
> **Response to Reviewer XVjC (1)**
>
> We thank the reviewer for his time and his appreciation of our work. Specifically, we are glad that the reviewer found our key technical results interesting, and our proofs clear and easy to follow. We also appreciate the fact that the reviewer describes the application of our technical results to the challenging problem of structured pruning as novel and worthy of publication.
>
> Furthermore, we would like to specifically highlight the following comments.
>
> **Originality and significance: While the topic of pruning has existed within the literature, the paper presents interesting contributions, both theoretically and empirically, that were shown to improve upon state of the art. Judging from this, the paper presents good originality and significance, and the topic is relevant to ICLR.**
>
> We thank the reviewer for the appreciation of the originality of our work!
>
> **Quality and clarity: Overall, the quality and clarity of the paper is decent, but can be improved with further revision. The paper effectively guides the reader via usage of questions and discussions. There are some issues surrounding the formatting of references, typos, etc., though. More suggestions regarding quality and clarity are presented in the following section.**
>
> We thank the reviewer for the positive review! We have tried to answer the specific points regarding the clarity of our work in the sequel.
>
> **Interesting-ness of results: The proofs and techniques presented within the paper is also interesting and insightful by itself. For example, the proofs stated in sections A.1 and A.2 in the appendix are simple, clean and to the point.**
>
> We sincerely thank the reviewer for the appreciation of the novel mathematical techniques introduced in this work.
>
> We now address the concerns raised by the reviewer.
>
> **Potential breach of anonymity: In Table 1, there is a column with header "TVSPrune(Murti et al., 2022) (our work)".**
>
> We apologize for this typo - "(our work)" was intended to be next to WitnessPrune on Table 1. We have amended this error.
>
> **Clarity issues: In certain parts of the paper, clarity issues cause confusion when reading. Some examples include:**
>
> **-   Formatting of references. The paper chooses to use references without separations from text. While this is suitable for some scenarios, in other situations it obstructs the flow of the reading, and it may be beneficial to update the references in these situations. This can likely be changed quite easily.**
> **-   Unclear definitions. There are multiple sections of the paper that can be revised to make definitions clearer. One example is in Section 2.2: the very first sentence abruptly ends with a "..., and let." Formatting issues and typos throughout the paper should be revised and updated.**
>
> We have updated the text with corrections to the main document. Please do check it out!
>
> **In Section 4.1, there is an assumption that that functions "g" and "G" exist. How strict is this assumption and how does it affect the main results?**
>
> The functions $g(\cdot)$ and $G(\cdot)$ need only exist implicitly. As noted in Theorems 1 and 2, we require that $\mathbb{E}\_{x\sim \mathbb{P}} [u^\top\varphi(X)] = g(\mathbf{P})$. Thus, so long as $\mathbb{E}\_{x\sim \mathbb{P}}[u^\top\varphi(X)]$ and $\mathbb{E}\_{x\sim \mathbb{P}}[(u^\top\varphi(X))^2]$ are finite, that is sufficient. This is also reflected in the proofs of Theorems 1 and 2, presented in Appendices A.1 and A.2 - the result only depends on the moments of the witness function $f(X)$ with respect to the distributions $\mathbb{P}$ and $\mathbb{Q}$.
>
> Moreover, this assumption is common. For Fisher Discriminant Analysis, it is assumed that the first two moments of the distributions exist, and given distributions $\mathbb{P}$ and $\mathbb{Q}$, we choose $\varphi(X) = X - \frac{1}{2}(\mu\_p - \mu\_q$, where $\mu\_p$ is the mean of distribution $\mathbb{P}$.
>
> **Are there theoretical justifications on whether WitnessPrune is generally preferable over previous methods such as TVSPrune?**
>
> Yes! TVSPrune[3]  makes a strong assumption that the class conditional distributions of the feature maps are spherical Gaussian. In this work, we make no such assumptions about the class-conditional distributions of the feature maps, immediately highlighting the improvement over works such as TVSPrune.
>
> **References**
>
> [1] *CHIP: CHannel Independence-based Pruning for Compact Neural Networks*. Sui et al, 2021.
>
> [2] https://github.com/Eclipsess/CHIP_NeurIPS2021 (CHIP github)
>
> [3] *Pruning Filters for Efficient Convnets*. Li et al, 2017.
>
> [4] *Provable Filter Pruning for Efficient Neural Networks*. Liebenwein et al, 2020.
>
> [5] *Dreaming to Distill: Data-free Knowledge Transfer via DeepInversion*. Yin et al, 2020.

---

> ### Author Response · Authors · 2023-11-17
> **Response to Reviewer XVjC (2)**
>
> **Regarding Table 1, why is CHIP unavailable for VGG19? Is it possible to produce the numbers for that experiment?**
>
> Unfortunately, CHIP[1] does not provide an implementation for VGG19, as seen in the main paper as well as the github repository [2], and as such, is untenable to implement at short notice. However, the authors provided an implemenation of ResNet50 trained on ImageNet. We are currently comparing our work with CHIP and TVSPrune on ResNet50 trained on ImageNet.
>
> **Also, regarding Table 1, how are the sparsity levels in the "Param. Sparsity" selected?**
>
> We do as follows:
> * For each layer $l$, we select a number of filters to be pruned, This is based on insights from [3],[4].
> * We then count the number of parameters removed when the filters are pruned.
> * We also count the number of kernels removed in layer $l+1$. For each filter pruned in layer $l$, a kernel ($3\times 3$ in VGGnets, and with various sizes for ResNets) is removed from every filter in layer $l+1$.
> * We write this formally as follows. Let $n_l$ be the number of filters in layer $l$,  let $r_l < n_l$ be the number of filters pruned in layer $l$, and let $L$ be the total number of layers. Let $K^2$ be the number of parameters in each convolutional kernel (as defined in . Note that $n_l$ is the number of input channels to layer $l+1$ in an unpruned model, and $n_l-r_l$ is the number of input channels to layer $l+1$ after pruning $r_l$ filters in layer $l$.
> * Thus, the number of parameters pruned is $N_{pruned} = \sum_{l=1}^{L}(r_{l}n_{l-1}K^2 ) + \sum_{l=2}^L(n_l-r_l)r_{l-1}K^2$. Here, $r_0=0$ and $n_0$ is the size of the input channels, which is 3 for RGB images.
> * For a concrete example, we consider the case of the VGG16 example stated in the second row of Table 1. Here, $L=13$. For $l=1,\cdots,7$, we let $r_l$ be the smallest integer greater than $0.1n_l$, and for $l=8,\cdots,13$, $r_l$ is the smallest integer greater than $0.6n_l$. For layers 8-13, this is $r_l=309$. Applying the formula, we get $N_{pruned}/N_{total} \approx .751$.
>
> **While there are some discussions on the variants of WitnessPrune in the appendix, it is not discussed thoroughly and the clarity there is lacking. Are there additional experiments to discuss the effectiveness of these variants?**
>
> We thank the reviewer for the feedback!  The key points we raise in this discussion are as follows.
>
> 1. We describe the different variants of WitnessPrune in greater detail, namely  WitnessPrune-FQ, WitnessPrune-M, and WitnessPrune-E. WitnessPrune-FQ uses the Fisher discriminant based lower bound described given in Corollary 1, only with quadratic as opposed to linear features,  to estimate the similarity between the distributions of feature maps, whereas WitnessPrune-M uses the Minimax Probability Machine (MPM) based bound stated in Corollary 2. WitnessPrune-E chooses the best separation of the two prior methods. Our experiments show that the Fisher discriminant based lower bound generally outperforms the MPM based lower bound on VGG and ResNet models trained on CIFAR10.
>
> 2. In Section C.3, we describe the use of the BatchNorm random variables (BNRVs). We define the BNRVs  in equation 15, and the moments of the BNRV are the BatchNorm running means and variances. We use this machinery to reduce the storage requirements of the moments, as they allow us to compute the moments of scalar functions, while still retaining sufficient distributional information, as noted in Yin et al [5].
>
> 3. We study the efficacy of the different variants of WitnessPrune by using each algorithm *without finetuning*, as that best illustrates the quality of the pruning algorithm.  Our experiments show that the WitnessPrune-FQ slightly outperforms WitnessPrune-M. We will present those empirical results within the next 24-48 hours.
>
>
> **References**
>
> [1] *CHIP: CHannel Independence-based Pruning for Compact Neural Networks*. Sui et al, 2021.
>
> [2] https://github.com/Eclipsess/CHIP_NeurIPS2021 (CHIP github)
>
> [3] *Pruning Filters for Efficient Convnets*. Li et al, 2017.
>
> [4] *Provable Filter Pruning for Efficient Neural Networks*. Liebenwein et al, 2020.
>
> [5] *Dreaming to Distill: Data-free Knowledge Transfer via DeepInversion*. Yin et al, 2020.

---

> ### Author Response · Authors · 2023-11-21
> **Response to reviewer XVjC - Additional Experiments**
>
> ### **Are there additional experiments to discuss the effectiveness of these variants?**
>
> We conducted an experiment comparing different variants of WitnessPrune. We focus on the VGG16 model trained on CIFAR10, as it is illustrative of this. We prune only the final 6 convolutional layers, each by a factor of .5 (so 256 filters are pruned from each layer). We compare 3 variants - WitnessPrune-FQ (Algorithm 2), WitnessPrune-MQ (Algorithm 3), and WitnessPrune-E (Algorithm 4), and the derivation of TVSPrune described in Appendix C. To show the utility of the different algorithms, we compare them without fine-tuning, and with 30 epochs of fine-tuning. We present the results in the Table below:
>
> |**Algorithm** |**WitnessPrune-FQ** |**WitnessPrune-MQ** | **WitnessPrune-E** | **TVSPrune**| WitnessPrune |
> |---|---|---|---|---|---|
> | **Accuracy drop** (No Fine-tuning)|  -8.54%|  -8.77%	| -8.54% 	| -8.62% | -7.85% |
> | **Accuracy drop** With Fine Tuning|  -1.21%|  -1.19%	|  -1.16%	| -1.08% | -1.03%|
>
> **Observations**
>
> On this model and dataset, WitnessPrune-FQ outperforms TVSPrune and WitnessPrune-MQ. WitnessPrune-E is effectively WitnessPrune-FQ, as the MPM-based lower bound is weaker than the Fisher-based lower bound (as we demonstrated in section 7.2). This is reflected in the accuracy of the unpruned model - they are almost identical. Note that all three methods are weaker than WitnessPrune (Algorithm 1), whose main results are presented in Table 1 in Section 7.3.
>
>
> ### Resnet Experiments - ResNet50 on Imagenet
> As promised previously, we have evaluated our method on a ResNet50 model trained on Imagenet. We compare WitnessPrune with CHIP [1], TVSPrune (the variant discussed in Appendix C.3), and L1 [2]. We fine-tune for 120 epochs with a batch size of 128. We present our results below.
>
>
> | **Algorithm** | WitnessPrune | CHIP | **TVSPrune** | **L1** |
> |---|---|---|---|---|
> | **20.2% sparsity** (No FT) |  **-2.67%**|  -2.99%|  -2.85%| -6.45%|
> | **20.2% sparsity** (FT) |  **+.02%**	|  **+.01%**	|  -.01%	| -1.08% |
> | **40.2% sparsity** |  **-2.58%**	|  -2.74%	|  -2.77%	| -4.45% |
>
> **Observations**
>
> We see that WitnessPrune clearly outperforms several contemporary baselines on the ImageNet task on the ResNet50 architecture, demonstrating its broad applicability to classification tasks. At low sparsity levels in particular,
>
>
> **We hope we have addressed your concerns. If there are any further concerns or questions you have, we are eager to engage in further discussions.**
>
> ### References
>
> [1] *TVSPrune - Pruning Non-discriminative filters via Total Variation separability of intermediate representations without fine tuning*. Murti et al, 2023.
>
> [2] *CHIP: CHannel Independence-based Pruning for Compact Neural Networks*. Sui et al, 2021.
>
> [3] *Pruning Filters for Efficient Convnets*. Li et al, 2017.

---

### Official Review · Reviewer_1f4m · 2023-11-01

**Soundness:** 3 good
**Presentation:** 3 good
**Contribution:** 2 fair
**Rating:** 3
**Confidence:** 3

**Summary:**

This paper presents a simple lower bound on the total variation (TV) distance between two distributions, and uses it to design novel pruning algorithms. To illustrate the basic idea in neural network pruning algorithms, consider a binary classification setup and class-conditional “feature” distributions (i.e., distribution of activations for each layer in the neural network). The approach considers a neuron “uninformative” if the TV distance between its conditional distribution for label 0 and label 1 is small, and prunes it. TV distance between high-dimensional distributions is in general difficult to compute, and the fact that we usually only have samples, not the densities, makes this issue even worse. The authors claim that their simple lower bound provides a more “tractable” alternative, though it is not clear what they mean by “tractable” in this context.

**Strengths:**

Overall, the paper is well-written. The ideas are clearly presented and I appreciate the simplicity of the TV lower bound and the straightforward application to pruning.

**Weaknesses:**

The paper leaves a lot of issues, both theoretical and practical, unaddressed. Thus, I am inclined to reject this paper at the moment, but I do see substantial potential in it should the authors address the issues mentioned below.

The most glaring issue is that the TV lower bound is stated only in terms of populational quantities, such as exact moments of the distribution. In practice, these quantities have to be *estimated* from finitely many samples. The regime the authors seem to have in mind is the fixed-dimension infinite-samples setting. However, for neural network pruning, their main application, a *high-dimensional* scaling in which the data dimension and the number of samples grow proportionally would be a more pertinent setting to analyze. This greatly complicates the analysis for their lower bound since one would need to analyze the estimation error in their moment functional. Even the simple task of estimating moment tensors of high-dimensional distributions can be tricky. Hence, a finite sample analysis of their lower bound is warranted.

Another issue, which is mostly of theoretical interest, is that the lower bound can be extremely loose in some cases. It would be helpful if the authors could provide some insights on the weakness of their lower bounds to provide a more balanced perspective. For example, it is well-known that for any positive integer k, there exists a discrete distribution that exactly matches the first k moments of the standard Gaussian (see e.g., [DKS17]). If P is this moment matching discrete distribution and Q is   the standard Gaussian, then TV(P,Q) = 1 but the lower bound is 0.

**References**
- [DKS17]: Ilias Diakonikolas, Daniel M. Kane, Alistair Stewart. Statistical Query Lower Bounds for Robust Estimation of High-dimensional Gaussians and Gaussian Mixtures. *FOCS* 2017.

**Questions:**

- Can the authors explain why a “witness function” is named as such? My guess is that it comes from the general form of the lower bound from A.1 and the authors view it as the function that achieves the supremum (a “witness” of the supremum), but it would be helpful to explain the term in the introduction.
- The use of descriptor “robust” when they say “robust lower bound” is somewhat misleading. “Robust” is typically used in the context of estimation, but there is no estimation analysis in this paper.
- Can \varphi be non-polynomial? The defining condition for \varphi: R^d \to R^n is that its expectation is a function of only the first k moments. Are there any non-polynomial functions that satisfy this condition?
- p.6 saliency score r_j^\ell subscript and superscript are reversed before Eq.(7).

---

> ### Author Response · Authors · 2023-11-17
> **Response to Reviewer 14fm (1)**
>
> We would like to thank the author for his appreciation of our work, and his insightful questions. In particular, we thank the reviewer for his appreciation of the main ideas presented in our work.
>
>
> **Overall, the paper is well-written. The ideas are clearly presented and I appreciate the simplicity of the TV lower bound and the straightforward application to pruning.**
> We thank the reviewer for the appreciation of the new results proposed in our work, as well as their application to neural network pruning.
>
> We would then like to address the reviewer's concerns.
>
> **The authors claim that their simple lower bound provides a more “tractable” alternative, though it is not clear what they mean by “tractable” in this context.**
> Typically, the approximating the TV distance is known to be  #P complete [9]. Our work is more tractable than prior art since it requires only the computation of the first two moments of a witness function $f(X) = u^\top \varphi(X)$, which, as stated in Section 4, are assumed to be functions of the moments of the distributions themselves, as noted in Section 4 of the main document.
>
> **The most glaring issue is that the TV lower bound is stated only in terms of populational quantities, such as exact moments of the distribution. In practice, these quantities have to be estimated from finitely many samples. The regime the authors seem to have in mind is the fixed-dimension infinite-samples setting.**
> We would like to clarify that in the setting of this paper - that is, using the TV-separation between the class-condtional distributions of the **feature maps generated by individual filters** as defined in Equation (1) on page 2-  the dimensions of feature maps, particularly in later layers, are not high-dimensional. That is, the number of samples of each class is actually greater than the dimension of the individual feature maps.  For models trained on CIFAR10, we typically choose 100 samples per class to compute the moments of the witness function $u^\top \varphi(X)$.
>
> **However, for neural network pruning, their main application, a high-dimensional scaling in which the data dimension and the number of samples grow proportionally would be a more pertinent setting to analyze.**
> In the final layers, of most neural network architectures, the size of the feature maps is often *low-dimensional*, and moreover, these layers have been shown to be most amenable to pruning [1],[7]. We illustrate this with a VGG-16 model trained on CIFAR10, where $n=100$ is the number of samples. We see that the dimension of the feature maps is relatively small.
>
>
> |Layers | 4,5,6 | 7,8,9 | 10,11,12 |
> |---|---|---|---|
> |  Dimension of feature map $d$ |  64 |  16 | 4  |
> | Ratio of dimension to sample size $d/n$ |  .64 |  .16 |  .04 |
>
> We see from this table that the majority of feature maps that can be effectively pruned are *not* high dimensional. Note that this trend can be observed over a variety of architectures as well.
>
> **This greatly complicates the analysis for their lower bound since one would need to analyze the estimation error in their moment functional. Even the simple task of estimating moment tensors of high-dimensional distributions can be tricky. Hence, a finite sample analysis of their lower bound is warranted.**
>
> For feature maps that are high dimensional, we note that in Corollaries 1 and 2, we derive lower bounds on the TV distance with respect to the Fisher Linear Discriminant [5] and the Minimax Probability Machine (MPM) [6].  We also recall that works such as [1],[7] observed that initial layers possessing high-dimensional feature maps are typically the most difficult to prune effectively. In this case, we can use existing methodologies, such as those proposed in  [8],[6] for robust estimation of Fisher and MPM based lower bounds on the TV distance. Specifically, if we choose $u^\top \varphi(X) = u^\top X$, we can directly apply Equation 15 proposed in *Robust Fisher Discriminant Analysis*, by Kim et al [8].
>
>  However, we feel that a broader investigation of these aspects is beyond the scope of our current work, which focuses on using the lower bounds for pruning, which it does effectively, as evidenced by the empirical results presented in Section 7.3.
>
> **References**
>
> [1] *Provable Filter Pruning for Efficient Neural Networks*. Liebenwein et al, 2020.
>
> [2] *A Kernel Two-Sample Test*. Gretton et al, 2012.
>
> [3] *A Witness Function Based Construction of Discriminative Models Using Hermite Polynomials*. Mhaskar et al, 2020.
>
> [4] *A Witness Two-Sample Test*. Kubler et al, 2022.
>
> [5] *Pattern Classification*. Duda and Hart, 2002.
>
> [6] *A Robust Minimax Approach to Classification*. Lanckriet et al, 2002.
>
> [7] *TVSPrune - Pruning Non-discriminative filters via Total Variation separability of intermediate representations without fine tuning*. Murti et al, 2023.
>
> [8] *Robust Fisher Discriminant Analysis*. Kim et al, 2005.
>
> [9] *On Approximating Total Variation Distance* Bhattaharyya et al, 2022.

---

> ### Author Response · Authors · 2023-11-17
> **Response to Reviewer 14fm (2)**
>
> **Another issue, which is mostly of theoretical interest, is that the lower bound can be extremely loose in some cases. It would be helpful if the authors could provide some insights on the weakness of their lower bounds to provide a more balanced perspective.**
>
> We thank the reviewer for this insightful and important question.
>
> * The lower bounds on TV distance proposed in this work are not tight for arbitrary distributions. As stated in Theorem 1 (Equation 3),
> $$\min\_{ \mathbb{P}\in \mathcal{S}\_{k}(\mathbf{P}),\mathbb{Q}\in \mathcal{S}\_{k}(\mathbf{Q})} \mathrm{TV} (\mathbb{P},\mathbb{Q}) \geq
>         \sup\_{u\in\mathbb{R}^n}\frac{\left(u^\top(g(\mathbf{P})-g(\mathbf{Q}))\right)^2}{2u^\top(G(\mathbf{P}) + G(\mathbf{Q}))u},$$ where $\mathcal{S}\_k(\mathbf{P})$ is the set of all distributions with moments $\mathbf{P}$, the lower bound is in the setting with *limited information*, like finite collections of moments.  Thus, given two finite collections of moments $\mathbf{P}$ and $\mathbf{Q}$ of the two distributions, Theorem 1 provides a *worst case* lower bound on the TV distance between *any* distributions with moments given by $\mathbf{P},$ $\mathbf{Q}$, as stated in Equation 3. Thus, the lower bound may be loose.
> * However, the purpose of the lower bounds proposed in this work is to determine the well-separatedness of the class conditional distributions of the feature maps generated by individual filters.  For this use case, it is sufficient to check whether the lower bound provided by Theorem 3 is high enough.
> * Another insight is provided in Section 7.1. We show the quality of the lower bound improves with a richer class of witness function. In particular, we show that with polynomial witness functions, higher-degree polynomials provide higher lower bounds than lower-degree polynomials, as they incorporate more distributional information in terms of higher-order moments.
> * Furthermore, there are cases where the true TV distance can be computed from the lower bound. For instance, when  the distributions $\mathbb{P}$ and $\mathbb{Q}$ are Gaussians with the same variance $\Sigma$, and with means $\mu_p$ and $\mu_q$, and if we choose $\varphi(X) = X - \frac{1}{2}(\mu_p-\mu_q)$, we can find the true TV distance by computing $u^*$ in equation 3. We also take this opportunity to point out that this provides a different perspective on discriminative ability - we consider a filter to be discriminative if the optimal classifier trained upon the features it generates has a low classification error. We detail this further in Appendix B3, in the amended draft.
>
> **For example, it is well-known that for any positive integer k, there exists a discrete distribution that exactly matches the first k moments of the standard Gaussian (see e.g., [DKS17]). If P is this moment matching discrete distribution and Q is the standard Gaussian, then TV(P,Q) = 1 but the lower bound is 0.**
>
> The lower bounds proposed in this work are *worst-case* lower bounds for the TV distance - that is, given a set of distributions $\mathcal{S}_k(\mathbf{M}_1)$ with moments $\mathbf{M}_1$, and a set of distributions $\mathcal{S}_k(\mathbf{M}\_2)$ with moments $\mathbf{M}_2$, Theorem 1 (and Equation 3 in particular) provides a lower bound for the minimum TV distance between any distribution in $\mathbb{P}\in \mathcal{S}_k(\mathbf{M}_1)$ and any  $\mathbb{Q}\in\mathcal{S}_k(\mathbf{M}_2)$.
>
> To illustrate this, suppose $\mathbf{M}_1 := (\mu_1,\Sigma_1)$ and $\mathbf{M}_2 := (\mu_2,\Sigma_2)$. Then $\mathcal{S}_k(\mathbf{M_1})$ is the set of *all* distributions with means and variances given by $\mathbf{M}_1$, including the normal distribution. The same is true for  $\mathcal{S}_k(\mathbf{M_2})$. Thus, Theorem 1 provides a lower bound between any two distributions with means and variances given by $\mathbf{M}_1$ and $\mathbf{M}_2$ respectively. Thus, if $\mathbf{M}_1 = \mathbf{M_2}$, the lower bound will be 0.
>
> Also, in section 7.1, we consider an example where we compare two Gaussians, with the means of both distributions is $(0,0)$. As we incorporate higher-order moments by choosing witness functions that are higher-order polynomials, we obtain nontrivial lower bounds on the TV distance between them.
>
>
> **References**
>
> [1] *Provable Filter Pruning for Efficient Neural Networks*. Liebenwein et al, 2020.
>
> [2] *A Kernel Two-Sample Test*. Gretton et al, 2012.
>
> [3] *A Witness Function Based Construction of Discriminative Models Using Hermite Polynomials*. Mhaskar et al, 2020.
>
> [4] *A Witness Two-Sample Test*. Kubler et al, 2022.
>
> [5] *Pattern Classification*. Duda and Hart, 2002.
>
> [6] *A Robust Minimax Approach to Classification*. Lanckriet et al, 2002.
>
> [7] *TVSPrune - Pruning Non-discriminative filters via Total Variation separability of intermediate representations without fine tuning*. Murti et al, 2023.
>
> [8] *Robust Fisher Discriminant Analysis*. Kim et al, 2005.
>
> [9] *On Approximating Total Variation Distance* Bhattaharyya et al, 2022.

---

> ### Author Response · Authors · 2023-11-17
> **Response to Reviewer 14fm (3)**
>
> **Can the authors explain why a “witness function” is named as such? My guess is that it comes from the general form of the lower bound from A.1 and the authors view it as the function that achieves the supremum (a “witness” of the supremum), but it would be helpful to explain the term in the introduction.**
>
> The term "witness function" is a term commonly used in the literature to describe functions that "witness" the effect of distributions. Some recent and highly cited works that utilize this terminology are [2],[3],[4].
>
> **-   The use of descriptor “robust” when they say “robust lower bound” is somewhat misleading. “Robust” is typically used in the context of estimation, but there is no estimation analysis in this paper.**
>
> In this work, we use the term "robust" in the sense of *distributional robustness*. As stated in a previous response, the proposed bound is a worst-case lower bound. Thus, given any distributions $\mathcal{P}$ and $\mathcal{Q}$  with collections of moments **P** and **Q** respectively, Theorem 1 provides a worst-case lower bound on the TV distance between them.
>
> **-   Can \varphi be non-polynomial? The defining condition for \varphi: R^d \to R^n is that its expectation is a function of only the first k moments. Are there any non-polynomial functions that satisfy this condition?**
>
> $\varphi$ can indeed be non-polynomial. For instance, if we choose $u^\top \varphi(X)$ to be the moment generating function, there are distributions for which closed-form expressions of the MGF exist. An example of this would be the univariate Gaussian distribution $\mathcal{N}(\mu,\sigma)$, where we can simply choose $\varphi(X) =  \mathrm{exp}(\mu t + \frac{1}{2}\sigma^2 t^2)$, for a given real $t$.
>
>
> **-   p.6 saliency score r_j^\ell subscript and superscript are reversed before Eq.(7).**
>
> Thank you for pointing out this error - we have amended it.
>
> **References**
>
> [1] *Provable Filter Pruning for Efficient Neural Networks*. Liebenwein et al, 2020.
>
> [2] *A Kernel Two-Sample Test*. Gretton et al, 2012.
>
> [3] *A Witness Function Based Construction of Discriminative Models Using Hermite Polynomials*. Mhaskar et al, 2020.
>
> [4] *A Witness Two-Sample Test*. Kubler et al, 2022.
>
> [5] *Pattern Classification*. Duda and Hart, 2002.
>
> [6] *A Robust Minimax Approach to Classification*. Lanckriet et al, 2002.
>
> [7] *TVSPrune - Pruning Non-discriminative filters via Total Variation separability of intermediate representations without fine tuning*. Murti et al, 2023.
>
> [8] *Robust Fisher Discriminant Analysis*. Kim et al, 2005.
>
> [9] *On Approximating Total Variation Distance* Bhattaharyya et al, 2022.

---

> ### Author Response · Authors · 2023-11-21
> **Additional response to Reviewer**
>
> We hope we have sufficiently answered the questions and concerns raised by you. If there are additional concerns or thoughts, we are eager to engage in further discussions to try and answer them.

---

> ### Comment · Reviewer_1f4m · 2023-11-22
>
> I thank the authors for their effort in addressing my questions. However, I will maintain my initial score, as I believe essential issues within the scope of this work remain unaddressed. The paper does not lack merit, but it is somewhat incomplete.
>
> - **Estimation error *is* within the scope of this work.** According to the paper, the novelty of this work lies in dispensing with distributional assumptions on the distributions, e.g., the Gaussian assumption of [Murti et al., 2022]. However, going beyond the simplifying distributional assumptions entails accounting for a wider range of distributions, some of which may pathological, and providing guarantees of comparable nature. Otherwise, it would be difficult to argue for the novelty of the work since generalization would have come at the cost of rigor.\
> Under the Gaussian assumption, finite-sample analysis is a non-issue since we have tight control over the estimation error. However, in the general setting of this work, where distributions are identified only up to their moment sequence, we no longer have this luxury.\
> Bounds on the estimation error would be useful for practitioners too since it would provide guidance on how to choose their \varphi’s (e.g., how many moments should I inspect before pruning?)
>
> - **False sense of validity.** The language used in this paper gives a false sense of validity in the pruning procedure. What is obtained from samples is NOT the TV lower bound. The TV lower bound requires **exact** **moments** of the unknown distributions. What is obtained is an estimate of the lower bound, which achieves validity (as a high confidence bound) only after getting bounds on the estimation error.\
> Related to this issue is the first bullet point on p2: “The bounds require no prior knowledge of the distributions, apart from the boundedness of the distributions moments.” It is true that the bounds, which are functions of *exact* moments, do not require prior knowledge of the distributions. However, we do not know the exact moments. They need to be estimated and the estimation error for the lower bound would depend on specific properties of the unknown distribution.
>
> - **“Robust lower bound“ is a misnomer.** The TV lower bounds are agnostic to distributional properties other than the moments. As the authors mentioned, the lower bounds are worst-case in the sense that it takes a supremum over the set of all distributions sharing the same (truncated) moment sequence. This, in fact, makes the lower bound extremely *sensitive* to estimation error since any non-zero error in the estimated moments would place the distribution in a totally different set, and then take the supremum over this incorrect set. Thus, calling it a “robust” lower bound is somewhat misleading.
>
> - **Unnecessarily general definitions.** The MGF example the authors gave for a non-polynomial test function \varphi seems incorrect. The fact that the MGF of a Gaussian can be expressed in terms of its first two moments comes from the fact that its **entire moment sequence** can be characterized by its first two moments. If we applied the same \varphi (i.e., the MGF) to say, a mixture of k Gaussians, it seems unlikely that it can be expressed in terms of finitely many moments (of the *mixture*, not its components). This shows that the current formulation of the test statistic is unnecessarily complicated and confusing. If there are no good examples of a non-polynomial \varphi, I suggest simply defining \varphi as a bounded-degree polynomial in X.

---

> > ### Author Response · Authors · 2023-11-23
> > **Additional Response to Reviewer 14fm**
> >
> > We would like to thank the reviewer for his insightful comments. Furthermore, we hope that we addressed the reviewer's concerns in the previous rebuttal.
> >
> > In particular, we addressed the following:
> > * **Looseness of the lower bound:** We agreed that the lower bound proposed in Theorem 1 is loose, However, we also illustrated cases from which the true lower bound could be recovered, specifically in **Corollary 4** in Section B.3.
> > * **Pruning setting:** We argued that the feature maps in CNNs  are not necessarily high dimensional, and illustrated that with the example of VGG16.
> > * **Distrubutions with the same truncated moment sequences:** We clarified that given two pairs of truncated moment sequences, at least one moment would need to be different in order to obtain a nontrivial lower bound.
> > * **Witness function name and typos:** We amended the typos pointed out by the reviewer, and noted that the phrase "witness function" is commonly used in the literature.
> >
> > We would also like to put forward the following arguments.
> > *  **General definition of $\varphi$:** We agree that finding examples of $\varphi$ that are not polynomials with bounded degree is difficult, As such, we will amend the manuscript to reflect this,
> > * **Use of 'robustness':** We agree that our lower bounds are not robust to estimation errors, and we'll amend our manuscript to reflect these changes as well.
> > * **Importance of estimation error and incompleteness of the work:** We agree that an investigation into estimation error is very important. However, we argue that:
> >   * Our current work focuses on the application of the new lower bound to structured pruning, and our experiments indicate that **the current method is highly effective at that task**. Moreover, we argue that our current work is sufficient in scope for publication, as in the literature, robustness analysis usually merits an entirely new investigation - see the robustness investigation of the Minimax Probability Machine [1] first provided in [2], and the Robust Fisher LDA paper [3].
> >   * Second, as the feature maps of CNNs, particularly in later layers, are typically not high dimensional, the effect of estimation error on network pruning is diminished. This is further highlighted when the Batchnorm Random variables are used instead of the feature maps (inspired by [4]), as they are scalars.
> >   * Last, we will add an additional discussion section to the manuscript, highlighting the need for a deeper investigation into moment estimation errors and the robustness of the proposed techniques.
> >
> > ### References
> >
> > [1] *Minimax Probability Machine*. Lanckriet et al, 2002.
> >
> > [2] *Robust Minimax Probability Machine*. Lanckriet et al, 2003.
> >
> > [3] *Robust Fisher Discriminant Analysis*. Kim and Boyd, 2005.
> >
> > [4] *Dreaming to Distill: Data-free Knowledge Transfer via DeepInversion*, Yin et al, 2020.

---

### Author Response · Authors · 2023-11-21
**General response to Reviewers**

We'd like to thank all the reviewers and the area chair for the time and effort put into reviewing our work. **We hope we've addressed the reviewers' concerns, and are excited to engage further with reviewers and gain further feedback on our work.**

We would first like to sincerely thank all the reviewers for their appreciation of and praise for our work. In particular,
* **Reviewers appreciated the quality of the presentation of ideas.**

 *  **Reviewers appreciated the key technical results, as well as their application to pruning.**

Reviewers also raised some common concerns.
* **Typos can affect the clarity of writing**

  * We have endeavoured to address these issues in the amended draft (now uploaded). We hope this addresses the reviewers' concerns.**
* **Experimental Validation could be better (noted by reviewers XVjC and G7wT)**
  * We present a snapshot of our experiments on ImageNet below. Our method achieves superior or comparable performance to several contemporary works.  Further experimental details can be found in the responses to **Reviewer XVjC** and **Reviewer G7wT**.

|**Algorithm** |WitnessPrune | CHIP  | **TVSPrune** | **L1** |
|---|---|---|---|---|
| **40.2% sparsity** (with fine-tuning) |  **-2.58%**	|  -2.74%	|  -2.77%	| -4.45% |

**List of Changes**

We also take this opportunity to list the changes made to the draft.
* Section 2 has been edited, and typos removed.
* Error in Table 1 pointed out by Reviewer XVjC has been fixed
* Typo pointed out by Reviewer 1f4m has been fixed
* In Appendix B.3, Corollary 4 has been introduced, to highlight cases when our lower bound can be used to compute the true TV distance (also in the response to reviewer 14fm).
* Other typos/minor errors have been addressed.

---

### Meta-Review · Area_Chair_n37h · 2023-12-06

**Metareview:**

The paper proposes a new pruning algorithm for compressing neural networks. The idea is that for a given filter, if the distributions of output for different classes are close (in TV distance) then we can prune the filter. The TV distance is hard to compute so one needs to use other bounds that are easier to estimate. Previous works assume that the distributions are Gaussian and use bounds that are only valid for Gaussian. The current work uses the first two moments of the distribution to derive a bound. The experiments show good empirical performance that is better than alternatives, especially without further fine-tuning.

On the downside, the result is more general but does not take into account the error in estimating the moments of the distributions. This error/cost to reduce the error will only go up with larger models. The original paper only tested on VGG16 but in the discussion with the reviewers, some limited extra experiments for Resnet50 was done, showing improved performance compared with alternatives but essentially the same performance after extra fine-tuning.

**Justification For Why Not Higher Score:**

While two reviewers put it marginally above the threshold, one reviewer puts it significantly below due to the concern about estimation error for the moments. On the balance, it seems to be a good work that can be accepted some day but a bit incomplete in the current state.

**Justification For Why Not Lower Score:**

N/A

---

### Decision · Program_Chairs · 2024-01-16

Reject